# Gut microbe-targeted choline trimethylamine lyase inhibition improves obesity via rewiring of host circadian rhythms

Rebecca C Schugar[1,2†], Christy M Gliniak[1,2†], Lucas J Osborn[1,2†], William Massey[1,2], Naseer Sangwan[1,2], Anthony Horak[1,2], Rakhee Banerjee[1,2], Danny Orabi[1,2], Robert N Helsley[1,2], Amanda L Brown[1,2], Amy Burrows[1,2], Chelsea Finney[1,2], Kevin K Fung[1,2], Frederick M Allen[1,2], Daniel Ferguson[1,2], Anthony D Gromovsky[1,2], Chase Neumann[1,2], Kendall Cook[1,2], Amy McMillan[1,2], Jennifer A Buffa[1,2], James T Anderson[1,2], Margarete Mehrabian[3], Maryam Goudarzi[1,2,4], Belinda Willard[1,2,4], Tytus D Mak[5], Andrew R Armstrong[6], Garth Swanson[3], Ali Keshavarzian[6], Jose Carlos Garcia-Garcia[7], Zeneng Wang[1,2], Aldons J Lusis[3], Stanley L Hazen[1,2,8], Jonathan Mark Brown[1,2]*

[1]Department of Cardiovascular and Metabolic Sciences, Lerner Research Institute Cleveland Clinic, Cleveland, United States; [2]Center for Microbiome and Human Health, Lerner Research Institute, Cleveland Clinic, Cleveland, United States; [3]Departments of Medicine, Microbiology, and Human Genetics, University of California, Los Angeles, Los Angeles, United States; [4]Proteomics and Metabolomics Core, Lerner Research Institute, Cleveland Clinic, Cleveland, United States; [5]Mass Spectrometry Data Center, National Institute of Standards and Technology (NIST), Gaithersburg, United States; [6]Department of Internal Medicine, Division of Gastroenterology, Rush University Medical Center, Chicago, United States; [7]Life Sciences Transformative Platform Technologies, Procter & Gamble, Cincinnati, United States; [8]Department of Cardiovascular Medicine, Heart Vascular and Thoracic Institute, Cleveland Clinic, Cleveland, United States

*For correspondence:
brownm5@ccf.org

†These authors contributed equally to this work

**Abstract** Obesity has repeatedly been linked to reorganization of the gut microbiome, yet to this point obesity therapeutics have been targeted exclusively toward the human host. Here, we show that gut microbe-targeted inhibition of the trimethylamine N-oxide (TMAO) pathway protects mice against the metabolic disturbances associated with diet-induced obesity (DIO) or leptin deficiency (Lep$^{ob/ob}$). Small molecule inhibition of the gut microbial enzyme choline TMA-lyase (CutC) does not reduce food intake but is instead associated with alterations in the gut microbiome, improvement in glucose tolerance, and enhanced energy expenditure. We also show that gut microbial CutC inhibition is associated with reorganization of host circadian control of both phosphatidylcholine and energy metabolism. This study underscores the relationship between microbe and host metabolism and provides evidence that gut microbe-derived trimethylamine (TMA) is a key regulator of the host circadian clock. This work also demonstrates that gut microbe-targeted enzyme inhibitors have potential as anti-obesity therapeutics.

## Editor's evaluation

Schugar et al. present data on the effects of a small molecule inhibitor (iodomethylcholine, IMC) of bacterial choline metabolism. This extends prior work from this team of scientists to focus on obesity-related phenotypes. They report a decrease in body weight in a diet-induced obesity model accompanied by lower insulin and improved glucose control. Remarkably, they also observed phenotypes in the ob/ob (leptin-deficient) model, which has severe obesity. They go on to describe additional phenotypes in IMC treated and control mice: gut microbiota, gene expression, and metabolomics, with a focus on circadian rhythm. Taken together, these data support the potential therapeutic value of IMC for treating obesity and associated metabolic diseases.

## Introduction

There is a growing body of evidence that microbes residing in the human intestine represent a critical environmental factor that influences virtually all aspects of human health and disease (*Brown and Hazen, 2015*; *Sonnenburg and Bäckhed, 2016*). Our gut microbiome plays a central role in vital processes such as energy harvest from our diet, entraining our immune system during early life, xenobiotic metabolism, and the production of a diverse array of small molecule metabolites that are essential to human life (*Brown and Hazen, 2015*; *Sonnenburg and Bäckhed, 2016*). Although the microbiome field has uncovered many correlative relationships with human health and disease, there are few examples whereby alterations in the gut microbiome have been causally linked. One of the earliest relationships established between gut microbes and human disease was the link between alterations in gut microbial phyla (Bacteroidetes and Firmicutes) and obesity susceptibility, and the microbial transplantation studies revealing obesity susceptibility as a transmissible trait (*Bäckhed et al., 2004*; *Ley et al., 2005*; *Turnbaugh et al., 2006*; *Turnbaugh et al., 2009*). Specifically, obese mice harbor gut microbial communities with enhanced capacity to harvest energy from indigestible carbohydrates (*Turnbaugh et al., 2006*), and transplantation of either cecal contents or feces from either obese mice or humans is sufficient to promote obesity and related insulin resistance in germ-free mouse recipients (*Bäckhed et al., 2004*; *Ley et al., 2005*; *Turnbaugh et al., 2006*; *Turnbaugh et al., 2009*). There is also accumulating evidence that antibiotic exposure in early life can predispose children to become overweight or obese later in life (*Ajslev et al., 2011*), and antibiotic treatment in mice prior to weaning increases obesity and related insulin resistance in adulthood (*Cho et al., 2012*). It is interesting to note that nearly all diseases where obesity is a predisposing comorbidity (diabetes, liver disease, cardiovascular disease, hypertension, chronic kidney disease, and diverse cancers) have been shown to have a clear gut microbiome link (*Canfora et al., 2019*; *Aron-Wisnewsky et al., 2020*; *Brown and Hazen, 2018*; *Knauf et al., 2019*; *Marques et al., 2018*; *McQuade et al., 2019*).

Although there is now ample evidence that gut microbes play a contributory role in the development of obesity and related metabolic disorders, obesity-targeted drug discovery to this point has focused solely on targets encoded by the human genome. Our knowledge is rapidly expanding regarding what types of microbes are associated with obesity and related disorders, including the repertoire of microbe-associated molecule patterns they harbor and the vast array of metabolites that they produce. However, there are very few examples where this information has been leveraged into therapeutic strategies. The microbiome-targeted therapeutic field has primarily focused on either prebiotic or probiotic approaches, yet thus far these approaches have resulted in very modest or non-significant effects in obesity-related disorders in human studies (*Aron-Wisnewsky et al., 2019*; *Asgharian et al., 2020*; *Reijnders et al., 2016*; *Madjd et al., 2016*; *Koutnikova et al., 2019*). As an alternative microbiome-targeted approach, we and others have begun developing non-lethal selective small molecule inhibitors of bacterial enzymes with the hopes of reducing levels of disease-associated microbial metabolites (*Roberts et al., 2018*; *Wang et al., 2015*; *Gupta et al., 2020*; *Organ et al., 2020*; *Orman et al., 2019*). We have recently shown that small molecule inhibition of the gut microbial transformation of choline into trimethylamine (TMA), the initial and rate-limiting step in trimethylamine N-oxide (TMAO) generation (*Wang et al., 2011*), can significantly reduce atherosclerosis, thrombosis, and adverse ventricular and kidney remodeling in mice (*Roberts et al., 2018*; *Wang et al., 2015*; *Gupta et al., 2020*; *Organ et al., 2020*). Since 2011 the gut microbe-associated TMAO pathway has been associated with many human diseases associated with obesity including atherosclerosis (*Wang et al., 2011*; *Koeth et al., 2013*), thrombosis (*Zhu et al., 2016*; *Zhu et al., 2017*),

chronic kidney disease (*Bell et al., 1991*; *Tang et al., 2015*), heart failure (*Tang et al., 2014*; *Trøseid et al., 2015*), cancer (*Bae et al., 2014*; *Xu et al., 2015*), and diabetes (*Shan et al., 2017*; *Miao et al., 2015*). Given the fact that obesity is a comorbidity in all of these human diseases, here we set out to determine whether selective small molecule inhibition of the gut microbial choline transformation into TMA, a metabolic activity catalyzed by the microbial choline TMA lyase CutC (*Craciun and Balskus, 2012*), can protect against metabolic disturbance in preclinical mouse models of obesity. Although there is clear evidence that TMAO can directly impact cell signaling in host platelets and immune cells (*Zhu et al., 2016*; *Seldin et al., 2016*), it is still incompletely understood how the TMAO pathway is mechanistically linked to obesity-related cardiometabolic diseases. It is well appreciated that manipulation of the gut microbiome can impact obesity in part via rewiring of the host circadian clock, and reciprocally it has been shown that an intact circadian clock in the host can impact oscillatory behavior of gut microbiota (*Org et al., 2015*; *Seldin et al., 2016*; *Thaiss et al., 2014*; *Liang et al., 2015*). Here, we hypothesized that gut microbe-driven TMA production may serve as a high fat diet (HFD)-induced signal that integrates diet-microbe-host interactions to shape obesity-related circadian dysfunction and metabolic disturbance.

## Results
### Microbial choline TMA lyase inhibition protects mice from obesity development

To assess whether small molecule inhibition of gut microbial TMA production can protect mice from obesity, we treated mice with the non-lethal mechanism-based bacterial choline TMA lyase inhibitor iodomethylcholine (IMC) (*Roberts et al., 2018*). This small molecule inhibitor exhibits potent in vivo inhibition of the gut microbial choline TMA lyase enzyme CutC, lowering host plasma TMAO levels >90% (*Roberts et al., 2018*; *Gupta et al., 2020*; *Organ et al., 2020*). Designed as a suicide substrate mechanism-based inhibitor, past studies reveal that the vast majority of IMC is retained in bacteria and excreted in the feces with limited systemic exposure of the drug in the host (*Roberts et al., 2018*). When administered in an HFD, IMC effectively reduces levels of both the primary product of CutC TMA and the host liver-derived co-metabolite TMAO (*Figure 1a and b*). IMC was effective in blunting diet-induced obesity (DIO) (*Figure 1c*), without altering food intake (*Figure 1d*). DIO mice treated with IMC also showed improvements in glucose tolerance (*Figure 1e*), exhibited marked reductions in plasma insulin levels in fed mice but not under fasting conditions (*Figure 1f*) and had significant reductions in fat mass (*Figure 1—figure supplement 1c*,d). It is important to note that the ability of IMC to reduce adiposity was specific to high fat feeding and was not seen in low fat-fed mice (*Figure 1—figure supplement 1c*,d). Also, IMC treatment did not significantly alter intestinal fatty acid absorption or hepatic steatosis (*Figure 1—figure supplement 2a*,f and 3). Next, we administered IMC to hyperphagic leptin-deficient ($Lep^{ob/ob}$) mice in a chow-based diet (*Figure 1g–l*). As expected, IMC effectively lowered both TMA and TMAO levels in $Lep^{ob/ob}$ mice (*Figure 1g and h*). Quite unexpectedly, IMC dramatically protected $Lep^{ob/ob}$ mice from body weight gain, where some of the drug-treated mice were 15–20 g lighter than control $Lep^{ob/ob}$ mice (*Figure 1i*). Notably, IMC-driven protection from obesity was not associated with decreased food intake, which was paradoxically higher overall in IMC-treated mice (*Figure 1j*). Instead, IMC treatment promoted large increases in energy expenditure through both the light and dark cycles (*Figure 1l*, *Figure 1—figure supplement 4a-c*) Although IMC effectively reduced body weight and adiposity in $Lep^{ob/ob}$ mice, this was not associated with improvements in glucose tolerance (*Figure 1k*). Given these striking results in DIO and $Lep^{ob/ob}$ obesity progression models, we next wanted to determine if IMC could likewise provide obesity protection in mice with established obesity (i.e., a treatment regimen). To test this, we fed C57BL/6 J mice HFD for 6 weeks to allow for mice to reach obese conditions (body weights ranging from 35 to 40 g) and then initiated IMC treatment. Although IMC did not promote weight loss, it did effectively reduce the trajectory of weight gain and improved glucose tolerance in this obesity treatment model (*Figure 1—figure supplement 4d, f*). During the early phases of this obesity treatment study, IMC-treated mice again exhibited unexpected increased food intake compared to controls (*Figure 1—figure supplement 4e*). It is important to note that long-term IMC treatment did not promote either liver or renal toxicity, as shown by normal plasma enzyme levels as well as normal tissue histology in liver and adipose depots (*Figure 1—figure supplement 5a-e*). To more comprehensively understand

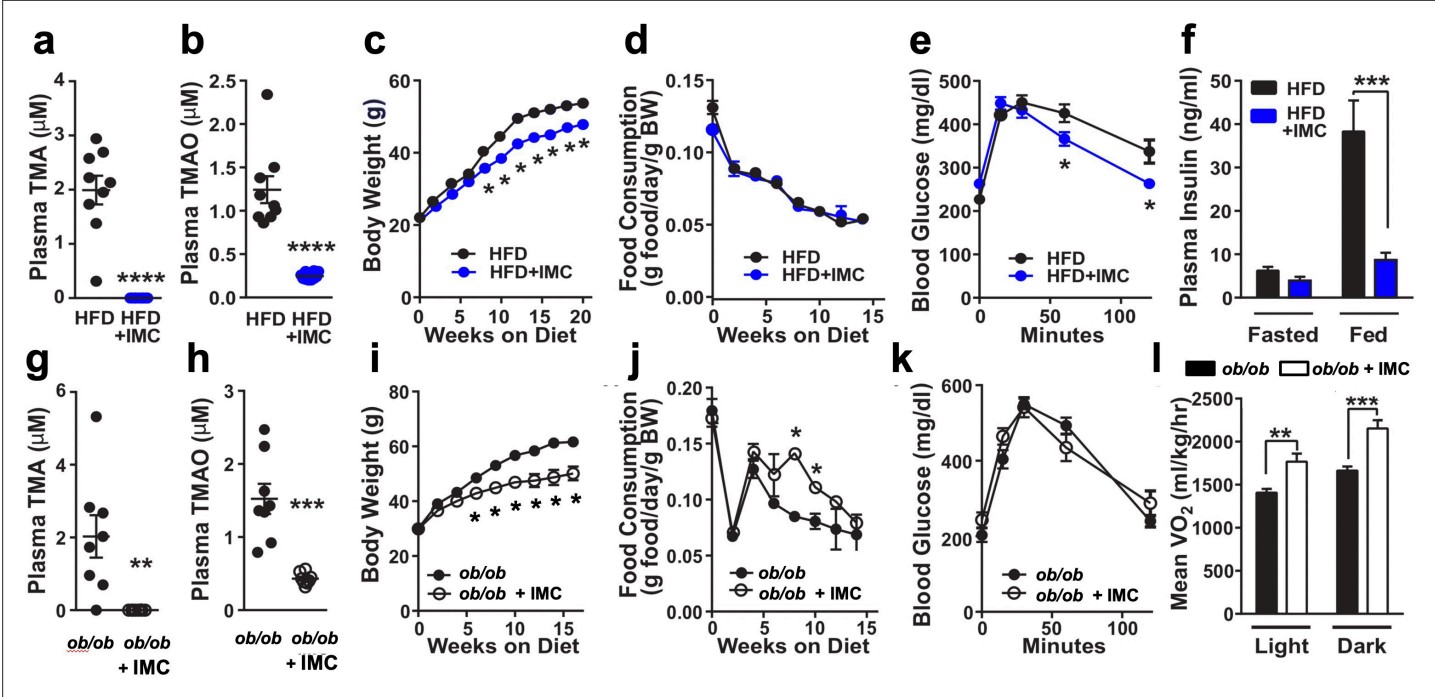

**Figure 1.** Small molecule choline trimethylamine (TMA) lyase inhibition improves obesity. Panels (**a–f**) and (**g–l**) represent data from control and iodomethylcholine (IMC)-treated high fat diet (HFD)-fed and *Lep^ob/ob* mice (chow-fed), respectively. (**a**) and (**g**) Plasma TMA levels. (**b**) and (**h**) Plasma trimethylamine N-oxide (TMAO) levels. (**c**) and (**i**) Biweekly body weights. (**d**) and (**j**) Biweekly food consumption. (**e**) and (**k**) Glucose tolerance test. (**f**) Plasma insulin levels. (**l**) Average oxygen consumption. In panels (**a–e**) and (**g–k**), groups were compared using t-tests. In panels (**f** and **l**), groups were compared using two-way analysis of variance (ANOVA) with Tukey's multiple comparisons test. Significance is defined as: *, $p < 0.05$. **, $p < 0.01$. ***, $p < 0.001$, ****, $p < 0.0001$. n = 6–10 per group.

The online version of this article includes the following figure supplement(s) for figure 1:

**Figure supplement 1.** Iodomethylcholine (IMC) does not confer antiobesogenic properties in a low fat diet (LFD) context.

**Figure supplement 2.** Iodomethylcholine (IMC) treatment does not alter hepatic steatosis in wildtype or genetically obese mice.

**Figure supplement 3.** Iodomethylcholine (IMC) treatment does not impact intestinal fat absorption.

**Figure supplement 4.** Choline trimethylamine (TMA) lyase inhibition increases energy expenditure and alters gene expression in white adipose tissue.

**Figure supplement 5.** Iodomethylcholine (IMC) supplementation does not lead to morphological changes in liver or adipose tissues and does not promote hepatic or renal toxicity.

the global effects of choline TMA lyase inhibition on white adipose tissue (WAT) gene expression, we performed unbiased RNA sequencing (*Figure 1—figure supplement 4g-j*). Principal coordinate analysis (PCA) of RNA expression profiles showed clear separation of adipose gene expression between HFD control vs. HFD with IMC (*Figure 1—figure supplement 4g*). The most differentially expressed genes altered by IMC were enriched in common pathways of adipogenesis, fatty acid metabolism, inflammation, and cytokine signaling (*Figure 1—figure supplement 4i*). Collectively, these data demonstrate that gut microbe-targeted choline TMA lyase inhibition can protect mice from obesity and selectively reorganize host adipose tissue gene expression.

## Choline TMA lyase inhibition facilitates alterations of the gut microbiome

One theoretical advantage of non-lethal microbe-targeted choline TMA lyase inhibitors, compared to antibiotic therapies, is that microbiome restructuring effects of the drug are expected to primarily target the species that rely on choline as a carbon or nitrogen source (*Maier et al., 2018*). Therefore, we next examined whether IMC treatment was associated with alterations in choline utilizers and other members of the gut microbiome community that may contribute to improvement in obesity-related metabolic disturbances. IMC treatment in HFD-fed mice resulted in significant restructuring of the cecal microbiome at every taxonomic level (*Figure 2a–e*). PCA of microbial taxa revealed distinct

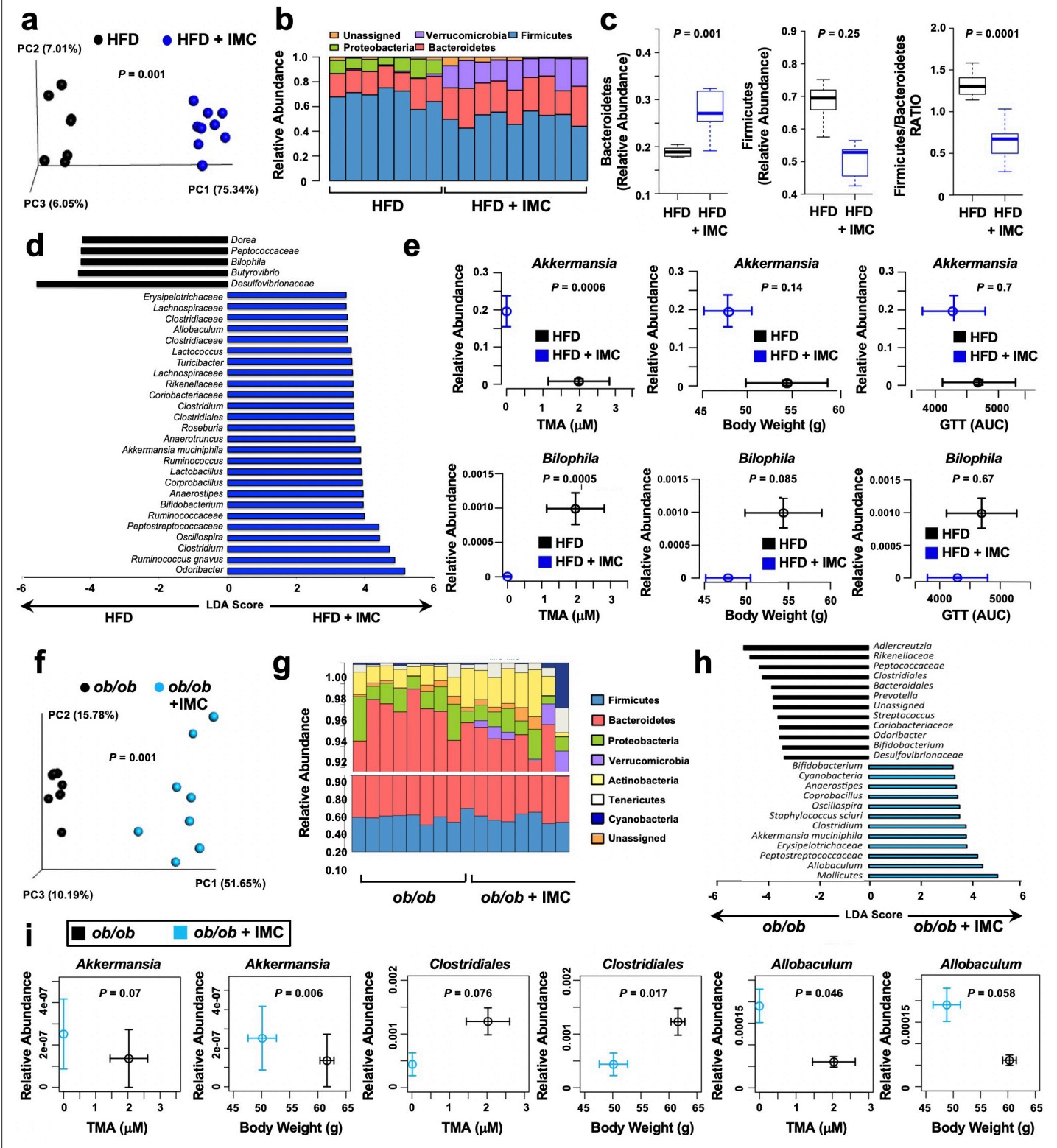

**Figure 2.** Small molecule trimethylamine (TMA) lyase inhibition alters the gut microbiome. Panels (**a–e**) and (**f–i**) represent data from control and iodomethylcholine (IMC)-treated high fat diet (HFD)-fed and *Lep^ob/ob* mice (chow-fed), respectively. (**a** and **f**) Principal coordinate analysis plot of microbiota profiles built from weighted Unifrac distances. Each point represents a single sample from a single mouse. Positions of points in space display dissimilarities in the microbiota, with points further from one another being more dissimilar. (**b** and **g**) Barplot of cecal microbiota at the phylum level. Each bar represents an individual mouse and each color an individual phylum. (**c**) Relative abundance of Firmicutes, Bacteroidetes, and the

*Figure 2 continued on next page*

*Figure 2 continued*

Firmicutes to Bacteroidetes ratio. The boxes represent the 25th and 75th quartiles, and the line displays the median value within each group. Points extending beyond the lines are outliers defined as values greater or less than 1.5 times the interquartile range. (**d** and **h**) Linear discriminatory analyses plot of taxa differing significantly with IMC treatment. (**e** and **i**) Correlation between taxa and plasma TMA levels, body weight, and glucose tolerance (HFD-fed mice only). Values in both X and Y directions are plotted as mean ± SEM.

The online version of this article includes the following figure supplement(s) for figure 2:

**Figure supplement 1.** Iodomethylcholine (IMC) supplementation leads to rapid kinetic changes in the gut microbiome community structure.

**Figure supplement 2.** Trimethylamine (TMA) lyase inhibition beneficially improves circadian oscillations in gut microbial communities.

clusters, indicating that IMC promoted restructuring of the cecal microbiome (*Figure 2a*). At the phylum level, HFD-fed IMC-treated mice had large increases in Verrucomicrobia and Bacteroidetes, and significant reductions in Firmicutes (*Figure 2b and c*). Of note, the ratio of Firmicutes to Bacteroidetes, which has been correlated with obesity in both humans and mice (*Bäckhed et al., 2004*; *Ley et al., 2005*; *Turnbaugh et al., 2006*; *Turnbaugh et al., 2009*), is significantly reduced by IMC treatment of the HFD-fed mice (*Figure 2c*). Performance of linear discriminant analysis coupled with effect size measurements (LEfSe analysis) revealed that IMC promoted significant reductions in the proportions of the taxa Desulfovibrionaceae, Butyrovibrio, Bilophia, Peptococcaceae, and Dorea, and increases in Odoribacter, Roseburia, and several members of the Clostridiaceae family (*Figure 2d*). We also examined whether the proportions of cecal genera were significantly correlated with plasma TMA, body weight, and glucose tolerance in DIO mice, and found that increases in *Akkermansia* and reductions in *Bilophila* were significantly correlated with plasma TMA level, and similar yet non-significant trends were found with body weight and glucose tolerance (*Figure 2e*). Remarkably, IMC rapidly reorganized the gut microbiome within 48 hr of the first drug exposure, and similar changes were seen after 6 days (*Figure 2—figure supplement 1a, b*) and a chronic treatment of 20 weeks (*Figure 2*). Similar to HFD-fed mice (*Figure 2a–e*), IMC also significantly altered the cecal microbiome in *Lep^{ob/ob}* mice (*Figure 2f-i*). In *Lep^{ob/ob}* mice, IMC significantly reduced the proportions of Adlercreutzia, Rikenellaceae, Peptococcaceae, and Clostridiales, while increasing levels of Molicutes, *Allobaculum*, Peptostreptococcaceae, and *Akkermansia* (*Figure 2g–h*). Correlation analysis in *Lep^{ob/ob}* mice revealed that IMC-induced alterations in *Akkermansia*, Clostridiales, and *Allobaculum* were significantly associated with circulating TMA and body weight (*Figure 2i*). Notably, in both the *Lep^{ob/ob}* and DIO models, the large increase in the proportions of Verrucomicrobia phylum seen with IMC are largely explained by increased level of *Akkermansia muciniphila*, which has been pursued as a therapeutic probiotic species for diabetes amelioration in other studies (*Everard et al., 2013*; *Shin et al., 2014*; *Org et al., 2015*) and is correlated to TMA and body weight phenotypes in the current study. Collectively, these data demonstrate that inhibition of gut microbial choline to TMA transformation with a selective non-lethal small molecule inhibitor promotes striking alterations of the gut microbiome that may contribute in part to improvements in energy metabolism and obesity observed in the host.

## The gut microbial TMAO pathway regulates host circadian rhythms

Although there is clear evidence that TMAO can directly impact cell signaling in macrophages (*Seldin et al., 2016*), endothelial cells (*Seldin et al., 2016*), and platelets (*Zhu et al., 2016*), it is still incompletely understood how the TMAO pathway is mechanistically linked to all of these common obesity-related diseases. Because of the recent appreciation of the gut microbiome as a coordinator of host metabolism and obesity-related phenotypes (*Org et al., 2015*; *Seldin et al., 2016*; *Thaiss et al., 2014*; *Liang et al., 2015*), we hypothesized that the metaorganismal TMAO pathway regulates host circadian rhythms to impact obesity and its associated cardiometabolic complications. The data that led us to this possibility includes: (1) misalignment of the host circadian clock is associated with the same human diseases that the TMAO pathway has been linked to *Org et al., 2015*; *Seldin et al., 2016*; *Thaiss et al., 2014*; *Liang et al., 2015*, (2) the gut microbiome oscillates in a circadian manner and this is highly sex specific (similar to the TMAO pathway) (*Org et al., 2015*; *Seldin et al., 2016*; *Thaiss et al., 2014*), and (3) disruption of the host circadian core clock machinery promotes gut microbial 'dysbiosis' and alters circulating levels of gut-derived metabolites (*Org et al., 2015*; *Seldin et al., 2016*; *Thaiss et al., 2014*; *Liang et al., 2015*). Given these clear overlaps, we investigated

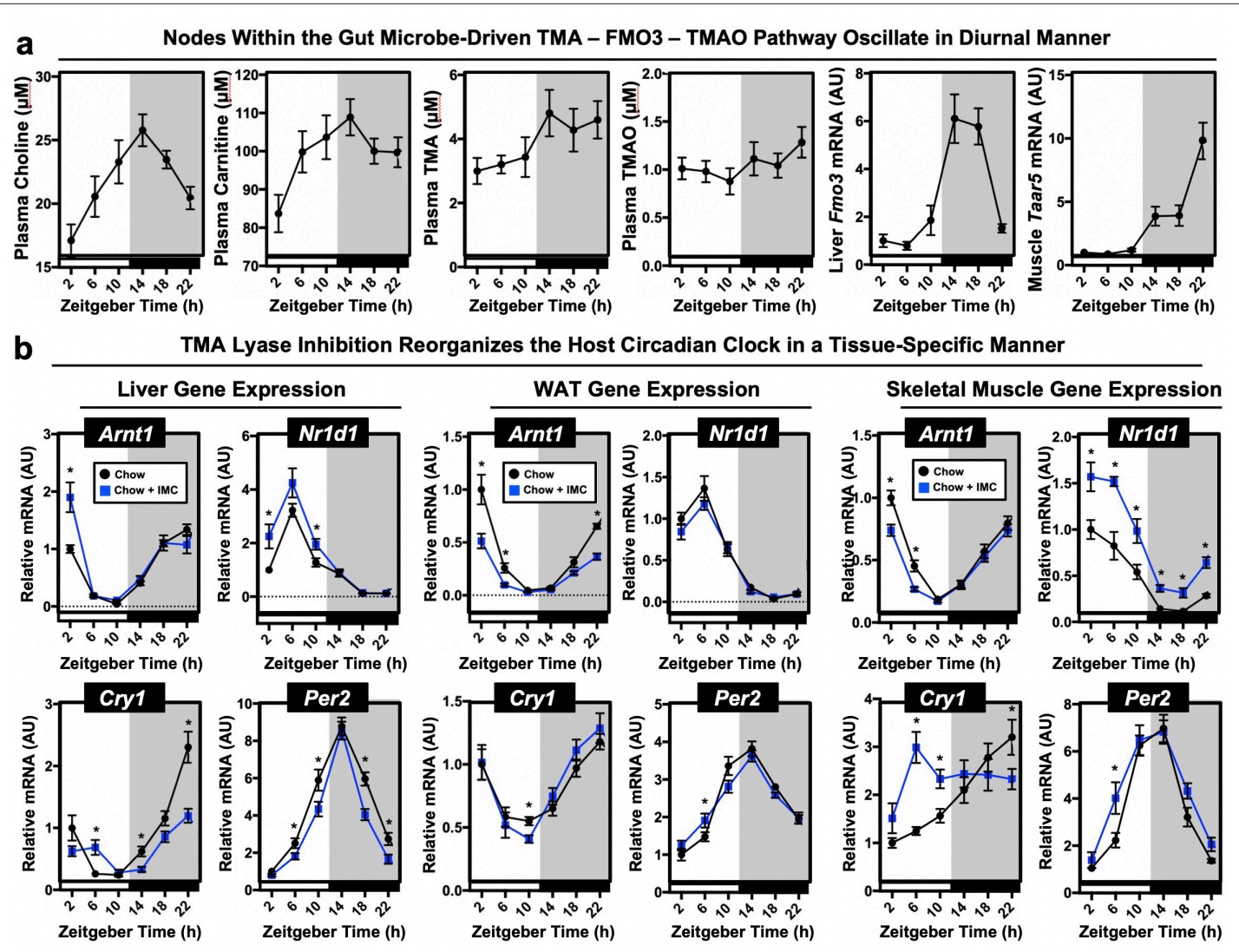

**Figure 3.** Nodes within the metaorganismal trimethylamine N-oxide (TMAO) pathway show diurnal rhythmicity and choline trimethylamine (TMA) lyase inhibition alters the host circadian clock. Wildtype male chow-fed C57BL/6J mice were necropsied at 4 hr intervals to collect plasma and tissue over a 24 hr period. (**a**) Plasma choline, L-carnitine, TMA, TMAO were quantified by liquid chromatography with online tandem mass spectrometry (LC-MS/MS); hepatic expression of *Fmo3* mRNA or gastrocnemius muscle expression of *Taar5* mRNA was quantified via quantitative PCR (qPCR) (n = 7–8). (**b**) Wildtype male C57BL/6J mice were fed chow or chow supplemented with the choline TMA lyase inhibitor iodomethylcholine (IMC) for 7 days. Mice were then necropsied at 4 hr intervals to collect liver, gonadal white adipose tissue, or gastrocnemius skeletal muscle. We then performed qPCR to examine the mRNA expression in liver, white adipose, and gastrocnemius skeletal muscle of key circadian clock regulators including aryl hydrocarbon receptor nuclear translocator like (*Arntl*; BMAL1), nuclear receptor subfamily one group D member 1 (*Nr1d1*; RevErbα), cryptochrome 1 (*Cry1*), or period 2 (*Per2*) (n = 7–8). Significance (p < 0.05) between diet groups at specified zeitgeber (ZT) time points was compared using Student's t-tests.

The online version of this article includes the following figure supplement(s) for figure 3:

**Figure supplement 1.** Trimethylamine (TMA) lyase inhibition alters the circadian rhythmicity of host metabolism-associated gene expression.

**Figure supplement 2.** Iodomethylcholine (IMC) treatment significantly alters the circadian oscillations in metabolic hormones and gene expression in high fat diet (HFD)-fed mice.

whether different substrates and regulatory genes within the TMAO pathway exhibited circadian rhythmicity. Indeed, we found that several nodes within the gut microbial TMAO pathway showed strong diurnal oscillations in mice (*Figure 3a*). Plasma levels of both choline, carnitine, and TMA were relatively low during the light cycle, and peak at the beginning of the dark cycle when mice are most active and consuming food (*Figure 3a*). However, TMAO showed more modest oscillatory behavior (*Figure 3a* and *Figure 3—figure supplement 1a*,d). We also found that the hepatic expression of

flavin-containing monooxygenase 3 (*Fmo3*) is quite low during the day but increases 6-fold during the early dark cycle (*Figure 3a*). However, this pattern for *Fmo3* expression is opposite in WAT (*Figure 4—figure supplement 1g*). Unexpectedly, we identified a tissue-specific circadian rhythms in the host TMA receptor (trace amine-associated receptor 5; *Taar5*) (*Figure 3a*). It is important to note that *Taar5* oscillation specifically occurs in skeletal (*Figure 3a*; *Figure 3—figure supplement 1c*,d; *Figure 4—figure supplement 1e*) and cardiac muscle (*Figure 4—figure supplement 1f*), and shows no oscillatory behavior in the olfactory bulb (*Figure 4—figure supplement 1g*), where it is important for sensing the 'fishy odor' smell of TMA (*Thaiss et al., 2016*; *Wang et al., 2017*). Collectively, these data suggest that the gut microbial TMAO pathway dynamically oscillates in a circadian manner.

## Gut microbe-targeted inhibition of TMA production rewires host-microbe metabolic crosstalk

To determine whether TMA or TMAO could potentially serve as a gut microbe-derived signal to entrain the host circadian clock, we treated a set of mice with IMC acutely (7 days) and examined effects on the core clock machinery as well as clock-mediated regulation of host metabolism. In this circadian study, IMC treatment effectively lowered circulating TMA and TMAO at every time point, yet there were still highly rhythmic low levels of plasma TMA and TMAO observed, possibly originating from microbial metabolism of non-choline (i.e., carnitine, γ-butyrobetaine, betaine, trimethyllysine) nutrient sources via microbial enzymes that are not inhibited by IMC (*Li et al., 2013*; *Wallrabenstein et al., 2013*; *Zhu et al., 2014*; *Figure 3—figure supplement 1a, d*). Unexpectedly, provision of IMC altered the expression of core clock transcription factors in a highly tissue-specific manner. In the liver, microbial choline TMA lyase inhibition modestly increased the peak amplitude of the core clock transcription factors *Bmal1* and *Nr1d1* yet blunted the rhythmic expression of *Cry1* (*Figure 3b*). In WAT, choline TMA lyase inhibition blunted the peak amplitude of *Bmal1* (*Figure 3b*). However, in skeletal muscle, we found the most striking differences in the core clock machinery where IMC-treated mice showed marked increases at all time points for *Nr1d1* and a near-complete phase shift for *Cry1* (*Figure 3b*; *Figure 3—figure supplement 1c, d*). Prior studies have confirmed that IMC is poorly absorbed in the host and selectively accumulates in gut microbiota (*Roberts et al., 2018*; *Gupta et al., 2020*). It is tempting to speculate that the circadian oscillations in skeletal muscle are highly drug responsive due in part to the fact that the TMA receptor *Taar5* oscillates there (*Figure 3a*; *Figure 3—figure supplement 1c, d*) to sense TMA and is not a 'direct' result of IMC per se, but rather an effect of IMC on gut microbial production of TMA. Critically, IMC does not alter the expression of core clock genes in areas of the central nervous system like the olfactory bulb where *Taar5* is abundantly expressed throughout the day and does not oscillate diurnally. Also, in mice fed a HFD to induce obesity, we found that IMC treatment significantly altered the oscillatory patterns of key host metabolic hormones insulin, leptin, and adiponectin (*Figure 3—figure supplement 2a-c*), which is likely to play a modulatory role in downstream lipid metabolic phenotypes. Furthermore, in HFD-fed mice IMC treatment altered the circadian oscillation in gene expression for cryptochrome 1 (*Cry1*) and period 2 (*Per2*) and key genes involved in lipid metabolism including carnitine palmitoyl transferase 1α (*Cpt1α*) and betaine homocysteine methyltransferase (*Bhmt*) (*Figure 3—figure supplement 2d-i*).

Recent literature has demonstrated that the composition of the gut microbiome is also under circadian regulation (*Org et al., 2015*; *Seldin et al., 2016*; *Thaiss et al., 2014*). These reports show that even at the phylum level, bacterial communities oscillate in a highly circadian manner, and gut microbe community oscillation can impact host metabolic rhythms that are linked to human disease (*Org et al., 2015*; *Seldin et al., 2016*; *Thaiss et al., 2014*). However, there is almost nothing known regarding what regulates this diurnal rhythm in gut microbial communities. Here, we demonstrate that microbial production of TMA, as modulated by the choline TMA lyase inhibitor IMC, has striking effects on circadian oscillations in gut microbial communities (*Figure 2—figure supplement 2a-c*). PCA revealed distinct group clusters, indicating that IMC promoted restructuring of the cecal microbial community composition in a circadian manner (*Figure 2—figure supplement 2a*). At the phylum level, IMC treatment results in large relative increases in Bacteroidetes, while lowering the overall levels of Firmicutes at all time points (*Figure 2—figure supplement 1e*, *Figure 2—figure supplement 2c*). Interestingly, IMC treatment promoted a significant increase in the abundance of *A. muciniphila*, proportions of which have been reported to be inversely correlated with body weight and insulin sensitivity in numerous studies in both rodents and humans (*Everard et al., 2013*; *Shin et al., 2014*; *Org et al.,*

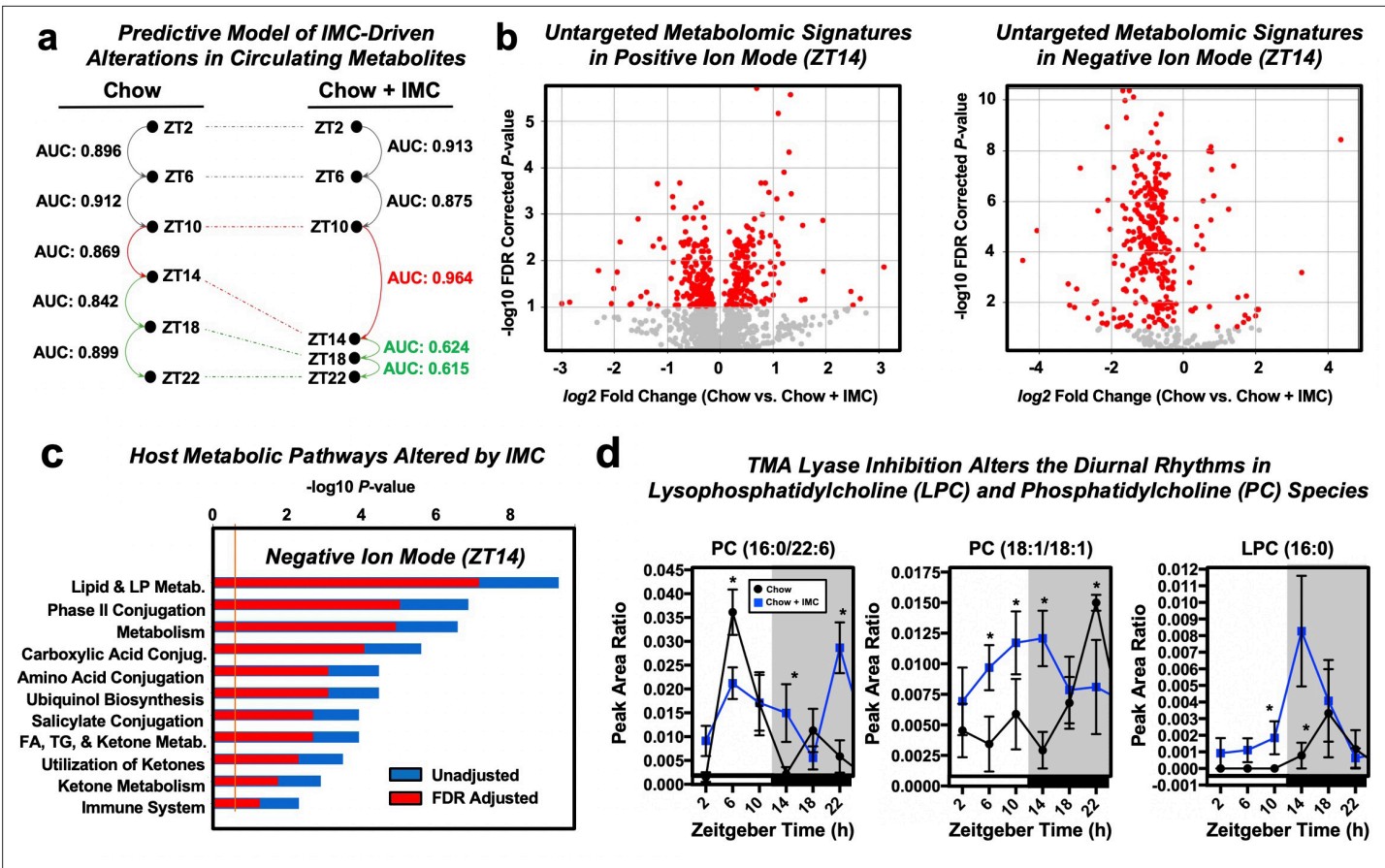

**Figure 4.** Small molecule inhibition of choline trimethylamine (TMA) lyases impacts circadian oscillations in choline-containing phospholipids in the host circulation. Wildtype male C57BL/6J mice were fed chow or chow supplemented with the choline TMA lyase inhibitor iodomethylcholine (IMC) for 7 days. Mice were then necropsied at 4 hr intervals to collect blood and tissue, and plasma time points were subjected to liquid chromatography with online tandem mass spectrometry (LC-MS/MS)-based untargeted metabolomics (n = 7–8). (**a**) Selective paired ion contrast analysis (SPICA) (*Lieber et al., 2019*) was used to identify time points where the plasma metabolome was most dramatically altered by IMC treatment compared to zeitgeber-matched controls. Pairwise analysis was conducted between all adjacent time points for the control and IMC treatment groups, resulting in a total of 10 comparisons made (five for Chow and five for Chow + IMC). Differences in the global plasma metabolome for each pairwise comparison were quantified via receiver operating characteristic (ROC) curve construction and area under the curve (AUC) calculations via Monte Carlo cross validation procedures in SPICA (*Lieber et al., 2019*) revealed that IMC most dramatically altered the plasma metabolome during the transition from ZT10 to ZT14. As a result, subsequent data analyses focused on this T10 to T14 transition, which also coincided with the light to dark phase transition and when the mice began to eat. (**b**) Volcano plots of the significantly altered metabolites (red) in positive and negative ion mode at the ZT14 time point. (**c**) Pathway analysis of data collected in negative ion mode at ZT14 reveals that IMC alters host lipid and lipoprotein metabolism among other metabolic pathways. (**d**) Relative plasma levels of phosphatidylcholine (PC) and lysophosphatidylcholine (LPC) species are altered by IMC.

The online version of this article includes the following figure supplement(s) for figure 4:

**Figure supplement 1.** Choline trimethylamine (TMA) lyase inhibition impact on the circadian rhythmicity of host hormone levels and gene expression.

*2015*; *Figure 2b, d, g and h*, *Figure 2—figure supplement 1c, d*, *Figure 2—figure supplement 2c*). In addition, IMC treatment increased the level of S24-7 Bacteroidetes family members, which have recently been reported to be depleted in several mouse models of obesity-related metabolic disturbance (*Koeth et al., 2014*; *Li et al., 2018*; *Lieber et al., 2019*; *Figure 2—figure supplement 2c*). Collectively, these data demonstrate that gut microbial production of TMA may be an important cue to impact the diurnal oscillations in gut microbial community structure and metabolic function in the host.

To understand how choline TMA lyase inhibition altered host metabolic processes in an unbiased way, we performed untargeted plasma metabolomics across a 24 hr period in IMC-treated mice (*Figure 4*). This large circadian metabolomics dataset was initially evaluated for any large-scale trends utilizing selective paired ion contrast analysis (SPICA) (*Liu et al., 2019*) to identify key zeitgeber (ZT)

time points where IMC elicited the most significant alterations in the plasma metabolome (*Figure 4a*). SPICA was utilized based on its unique ability to highlight subtle changes in the metabolome via analysis of metabolite pairs, which vastly reduces normalization issues endemic to mass spectrometry derived small molecule datasets (i.e., untargeted metabolomics). This approach delivers a more comprehensive picture of the overall metabolomic signature in TMA-regulated host circadian clock compared to the traditional single-metabolite oriented metabolomic data tools. Although there were clear IMC-induced metabolomic alterations at every time point, the most significant alterations were seen during the ZT10 to ZT14 transition, which coincides with the light to dark transition when mice typically begin to forage for food. Focusing on the ZT14 time point, we observed a large number of differentially abundant metabolites in the IMC-treated group that were in common pathways of lipid and lipoprotein metabolism, phase II conjugation, and other various macronutrient metabolic pathways (*Figure 4b and c*). In particular, we found that choline TMA lyase inhibition was associated with alterations in diurnal rhythms of phosphatidylcholine (PC) co-metabolites (*Figure 4d*). Gut microbial TMA lyase inhibition caused near-complete circadian phase shifts for two PC species (16:0/22:6 and 18:1/18:1), and two saturated species (16:0 and 18:0) of lysophosphatidylcholine (LPC), which showed robust oscillatory behavior distinct from control mice (*Figure 4d*). Furthermore, we found that the oscillatory behavior of genes involved in PC synthesis including choline kinase α (CKα) and phosphatidylethanolamine methyltransferase (PEMT) were significantly altered in TMA lyase inhibited mice (*Figure 3—figure supplement 1b, d*). It is important to note that IMC did not alter key plasma hormones (corticosterone or insulin) or lipids (non-esterified fatty acids) that are known to have strong effects on phospholipid metabolism themselves (*Figure 4—figure supplement 1a, b*). Collectively, these data suggest that gut microbial TMA production from the choline TMA lyase CutC is a critical determinant of host circadian rhythms in PC homeostasis.

## Choline TMA lyase inhibition-induced alterations in the gut microbiome reduce adiposity and alter host circadian rhythms

Given the consistent and profound effects of choline TMA lyase inhibition on the gut microbial community (*Figure 2*, *Figure 2—figure supplement 1*), we sought to determine if the observed phenotypes could be attributed to these changes by colonizing germ-free mice with cecal contents from HFD- or HFD + IMC-fed donor mice (*Figure 5a*). Gut microbial communities display circadian changes and microbes transplanted from mice receiving choline TMA lyase inhibitors results in significant taxonomic differences from mice receiving control microbiota (*Figure 5b–d*). Interestingly, the gut microbial community from HFD + IMC mice conferred reduced visceral adiposity to donor mice challenged with 8 weeks of HFD and these effects are independent of body weight, liver weight, and TMA/TMAO production (*Figure 5e–i*). Additionally, the transplanted cecal microbiota from HFD + IMC mice altered gene expression in a tissue-specific manner in alignment with our other studies (*Figure 3b*). Specifically, the expression of multiple circadian genes in the skeletal muscle, including *Nr1d1*, *Cry1*, *Cry2*, and *Per2*, was significantly reduced by the HFD + IMC microbiota at ZT14, the same time point that was highlighted in our unbiased metabolomics analysis (*Figure 5j*, *Figure 4*). In WAT, *Foxo1* expression was significantly increased at ZT14 vs. ZT2 only in mice that received an HFD cecal microbiota and in contrast *Adrb1* was increased at ZT14 in mice colonized with an HFD + IMC microbiota, but not those with the HFD gut microbes (*Figure 5k*). It is important to note that the cecal microbial transfer experiments into germ-free mice (*Figure 5*) were on a slightly different light cycle (14:10) compared to all other studies (12:12), which was necessitated by our gnotobiotic facility standard operating procedures. It remains possible that this lighting change may alter some of the circadian and metabolic phenotypes under study. Collectively, these findings further demonstrate the role of the gut microbiota in host physiology and as a therapeutic target through which pharmacological intervention may be deliberately directed.

## Discussion

Although obesity drug discovery has historically targeted pathways in the human host, a fertile period in biomedical research lies ahead where we instead target the microorganisms that live within us to improve obesity and related disorders. This paradigm shift in drug discovery is needed in light of the clear and reproducible associations between the gut microbiome and almost every human

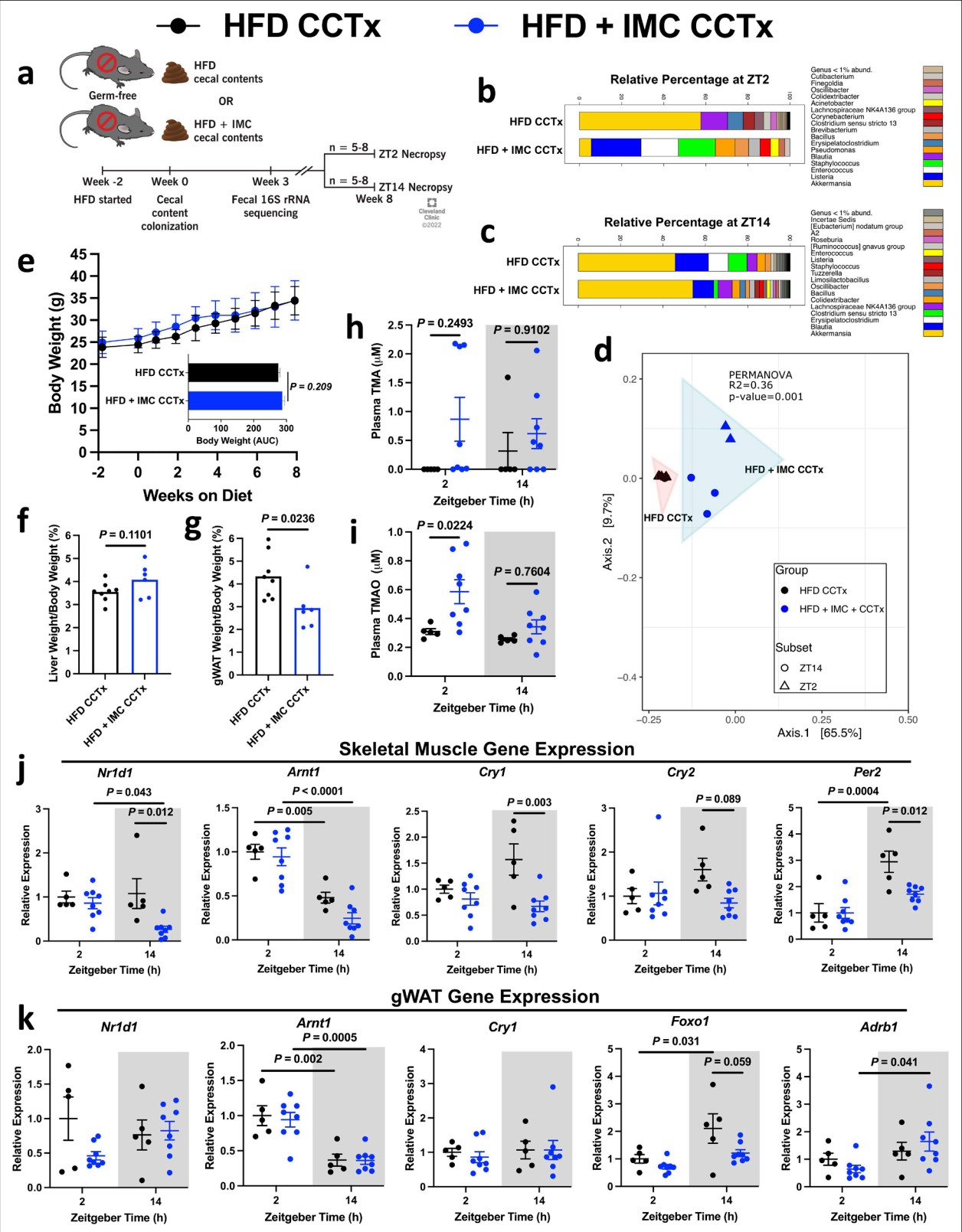

**Figure 5.** Cecal content transplant into germ-free mice informs adiposity and displays community-level circadian rhythmicity. Germ-free male B6/N mice were placed on a high fat diet (HFD) 2 weeks prior to cecal content gavage from mice fed either HFD or HFD + iodomethylcholine (IMC) for 6 days. (**a**) Schematic representation of experimental design. (**b**) Stacked bar chart representing circadian rhythmicity at ZT2 and (**c**) ZT14 within the HFD cecal contents transplant (HFD CCTx) and HFD + IMC cecal contents transplant (HFD + IMC CCTx) groups (n = 3–5). (**d**) Principal component analysis

*Figure 5 continued on next page*

*Figure 5 continued*

comparing HFD CCTx vs. HFD + IMC CCTx beta diversity of cecal 16S rRNA profiles at ZT2 and ZT14 8 weeks post-colonization (n = 2–5). (**e**) Weekly body weights with area under the curve (n = 8–10, error bars represent SD). (**f**) Liver and (**g**) gWAT weights as expressed as a percentage of total body weight (n = 6–8). (**h**) Plasma trimethylamine (TMA) and (**i**) trimethylamine N-oxide (TMAO) concentrations (n = 5–8). (**j**) Quantitative PCR (qPCR) analysis of key circadian rhythmicity genes in skeletal muscle (n = 5–8). (**k**) qPCR analysis of key circadian rhythmicity genes in gWAT (n = 5–8). Statistical analysis in panels (**e–g**) was performed using Student's two-tailed t-test. Statistical analysis in panels (**h–k**) was performed using two-way analysis of variance (ANOVA) with Tukey's multiple comparisons test.

disease. Now we are faced with both the challenge and opportunity to test whether microbe-targeted therapeutic strategies can improve health in the human metaorganism without negatively impacting the symbiotic relationships that have co-evolved. Although traditional microbiome manipulating approaches such as antibiotics, prebiotics, probiotics, and fecal microbial transplantation have shown their own unique strengths and weaknesses, each of these presents unique challenges particularly for use in chronic diseases such as obesity. As we move toward selective non-lethal small molecule therapeutics, the hope is to have exquisite target selectivity and limited systemic drug exposure given the targets are microbial in nature. This natural progression parallels the paradigm shifts in oncology which have transitioned from broadly cytotoxic chemotherapies to target-selective small molecule and biologics-based therapeutics. Here, we provide the first evidence that a selective non-lethal inhibitor to the microbial choline TMA lyase CutC can have advantageous effects on host energy metabolism, adiposity, and insulin sensitivity. The current study demonstrates that gut microbe-targeted suppression of choline to TMA generation: (1) protects the host from DIO and glucose intolerance, (2) protects *Lep^(ob/ob)* mice from obesity, (3) increases energy expenditure and alters the expression of lipid metabolic genes in WAT, and (4) rapidly reorganizes gut microbial communities to reduce the Firmicutes to Bacteroidetes ratio and increase abundance of *A. muciniphila*. This study also reveals that nodes within the gut microbial TMAO pathway exhibit circadian oscillatory behavior, and that inhibition of microbial choline to TMA transformation is associated with alterations in: (1) the expression of core clock machinery in metabolic tissues; (2) the diurnal rhythmic behavior of gut microbial communities; and (3) circadian oscillation in choline-containing phospholipids; (4) and that these observations are at least in part transferred in a gnotobiotic context. Collectively, our findings reinforce the notion that the gut microbial TMAO pathway is a strong candidate for therapeutic intervention across a spectrum of obesity-related diseases and have uncovered metaorganismal crosstalk between gut microbe-derived TMA and circadian metabolic oscillations in mice. Interestingly, recent meta-analysis of clinical studies demonstrated that circulating TMAO levels are dose-dependently associated with obesity in humans (*Moreira Júnior et al., 2019*), highlighting the potential translation of this work.

As drug discovery advances the area of small molecule non-lethal bacterial enzyme inhibitors it is key to understand how these drugs impact microbial ecology in the gut and other microenvironments. As we have previously reported (*Roberts et al., 2018*; *Gupta et al., 2020*; *Organ et al., 2020*), IMC treatment does promote a significant remodeling of the cecal microbiome in mice. These IMC-induced alterations occur as early as 1 week (*Figure 2—figure supplement 1b, c, e*) and persist even after 20 weeks (*Figure 2*). It is important to note that the observed IMC-induced alterations in the gut microbiome are generally expected to be favorable for the host. For instance, IMC lowers the ratio of Firmicutes to Bacteroidetes (*Figures 2c, 5b and c*, *Figure 2—figure supplement 1e*), which is known to be elevated in obese humans and rodents (*Bäckhed et al., 2004*; *Ley et al., 2005*; *Turnbaugh et al., 2006*; *Turnbaugh et al., 2009*). Of the taxa that were significantly correlated with circulating TMA levels (*Akkermansia*, *Bilophila*, Clostridiales, and *Allobaculum*), only *Akkermansia* was likewise correlated with body weight in *Lep^(ob/ob)* mice (*Figure 2i*). It is important to note that previous studies have shown favorable effects of *A. muciniphila* in obesity and diabetes, which has prompted its development as a probiotic (*Miao et al., 2015*; *Craciun and Balskus, 2012*; *Everard et al., 2013*). Supplementation with *A. muciniphila* can improve glucose tolerance in obese mice (*Everard et al., 2013*; *Shin et al., 2014*; *Org et al., 2015*). Furthermore, increases in *A. muciniphila* are consistently seen in mice treated with TMA lyase inhibitor (*Roberts et al., 2018*; *Gupta et al., 2020*; *Organ et al., 2020*), suggesting that small molecule inhibition of gut microbial choline TMA lyase may be an alternative means to enrich gut microbiomes with *A. muciniphila*. As small molecule bacterial enzymes inhibitors are developed, it will be extremely important to understand their effects on microbial ecology, and it is expected that some of the favorable effects of these drugs will indeed originate from the restructuring

of gut microbiome communities. This is not an uncommon mechanism by which host-targeted drugs impact human health. A recent study showed that nearly a quarter of commonly used host-targeted drugs have microbiome-altering properties (*Maier et al., 2018*), and in the context of diabetes therapeutics, it is important to note that metformin's anti-diabetic effects are partially mediated by the drug's microbiome-altering properties (*Dehghan et al., 2020*). Given the strong association between gut microbiome and obesity and diabetes (*Bäckhed et al., 2004*; *Ley et al., 2005*; *Turnbaugh et al., 2006*; *Turnbaugh et al., 2009*), it will likely be advantageous to find therapeutics that beneficially remodel the gut microbiome as well as engage either their microbe or host target of interest.

One burgeoning area of microbe-host crosstalk relevant to human disease is the intersection between gut microbes and host circadian rhythms (*Org et al., 2015*; *Seldin et al., 2016*; *Thaiss et al., 2014*; *Liang et al., 2015*). It is well appreciated that a transcriptional-translational feedback loop (TTFL) exists in mammalian cells to orchestrate an approximately 24 hr oscillatory rhythm in the expression of thousands of genes (*Org et al., 2015*; *Seldin et al., 2016*; *Thaiss et al., 2014*; *Liang et al., 2015*). The mammalian clock is coordinated by core transcription factors CLOCK and BMAL1, which peak during light phases, and crytochromes (CRYs) and period genes (PERs), which are most active during dark phases. Under normal conditions, the clock regulated TTFL maintains cell autonomous homeostatic responses to environmental 'zeitgebers' including light, food, xenobiotics, and exercise (*Org et al., 2015*; *Seldin et al., 2016*; *Thaiss et al., 2014*; *Liang et al., 2015*). However, disruption of normal circadian rhythms induced by abnormal light exposure, sleep-activity, or eating patterns has been associated with the development of many human diseases including obesity, diabetes, CVD, kidney disease, cancer, and neurological disease (*Org et al., 2015*; *Seldin et al., 2016*; *Thaiss et al., 2014*; *Liang et al., 2015*). Therefore, 'chronotherapies' or therapies that prevent circadian disruption hold promise across a wide spectrum of diseases. Interestingly, it has been discovered that circadian disruption is associated with a marked reorganization of gut microbial communities, and microbial abundance in the gut exhibits circadian rhythmicity (*Org et al., 2015*; *Seldin et al., 2016*; *Thaiss et al., 2014*; *Liang et al., 2015*). However, whether gut microbial metabolites contribute to circadian disruption is not well understood. Here, we show that multiple nodes with the metaorganismal TMAO pathway (choline, TMA, *Fmo3*, and *Taar5*) oscillate in a circadian manner, and that inhibition of the choline TMA lyase CutC rewires the host circadian clock itself, circadian rhythms in the cecal microbiome, and clock-driven reorganization of host lipid metabolic processes. The impact of gut microbial choline TMA lyase inhibition on the expression of core clock genes is modest in metabolic tissues such as the liver or WAT (*Figure 3b*; *Figure 3—figure supplement 1b, d*). However, in skeletal muscle where the host TMA receptor *Taar5* exhibits oscillatory behavior (*Figure 3a*), IMC treatment elicits profound alterations in the expression of *Nr1d1* (RevErbα) and cryptochrome 1 (*Cry1*) (*Figure 3b*). Furthermore, gnotobiotic mice receiving a cecal content transplant from IMC-treated mice had a marked suppression of *Cry1* and *Per2* in the skeletal muscle after 8 weeks (*Figure 5j*). The oscillatory behavior of the TMA receptor *Taar5* is specific to both skeletal and cardiac muscle, and is not seen in the olfactory bulb where *Taar5* is essential to sense the fish-like odor of TMA (*Thaiss et al., 2016*; *Wang et al., 2017*). A recent report demonstrated that TMAO, but not TMA, can bind to and activate the endoplasmic reticulum stress (ER stress)-related kinase PERK (*Eif2ak3*), and that TMAO binding to PERK regulates the expression of the forkhead transcription factor *Foxo1* (*Chen et al., 2019a*). It is interesting to note that PERK was recently shown to regulate circadian rhythms that support sleep/wake patterns and cancer growth (*Wu et al., 2017*; *Chen et al., 2019b*). We also observed tissue-specific alterations in the circadian expression of *Foxo1* with IMC treatment (*Figure 3—figure supplement 1b*,d; *Figure 4—figure supplement 1d*, *Figure 5k*). However, additional studies are needed to understand whether the TMAO-PERK-FOXO1 signaling axis is mechanistically linked to oscillatory behavior in metabolism or associated metabolic disease. Additionally, several independent metabolomics studies have found that circulating levels of TMAO exhibit circadian oscillatory behavior (*Ly et al., 2020*; *Bu et al., 2018*; *Beli et al., 2019*). In a similar manner, the expression of flavin-containing monooxygenase enzymes which convert TMA to TMAO are also under direct circadian clock transcriptional regulation (*Giskeødegård et al., 2015*; *Bollard et al., 2001*; *Dixit and Roche, 1984*). Collectively, these emerging data suggest that gut microbe-derived TMA and potentially its metabolic co-metabolite TMAO may be underappreciated microbe-derived signals that impact host circadian rhythms in metabolism. Further studies are now warranted to test whether TMA lyase

inhibitors of the host liver TMAO-producing enzyme FMO3 can act as chronotherapies to improve circadian misalignment-associated diseases.

The metaorganismal TMAO pathway represents only one of many microbial metabolic circuits that have been associated with human disease. Many microbe-associated metabolites such as short chain fatty acids, secondary bile acids, phenolic acids, and polyamines have been associated with many human diseases (*Brown and Hazen, 2015*; *Brown and Hazen, 2018*). In a circadian and/or meal-related manner, gut microbes produce a diverse array of metabolites that reach micromolar to milli-molar concentrations in the blood, making the collective gut microbiome an active endocrine organ (*Brown and Hazen, 2015*). Small molecule metabolites are well known to be mediators of signaling interactions in the host, and this work provides evidence that diet-microbe-host metabolic interplay can shape normal diurnal rhythms in metabolism and obesity susceptibility in mice. Our work, along with that of many others, demonstrates clear evidence of bi-directional crosstalk between the gut microbial endocrine organ and host metabolism. As drug discovery advances it will be important to move beyond targets based solely in the human host. This work highlights that non-lethal gut microbe-targeted enzyme inhibitors can serve as effective anti-obesity therapeutics in preclinical animal models and provides proof of concept of a generalizable approach to target metaorganismal crosstalk in other disease contexts. Selective mechanism-based inhibition of bacterial enzymes has the advantage over host targeting given that small molecules can be designed to avoid systemic absorption and exposure, thereby minimizing potential host off-target effects. As shown here with the gut microbial TMAO pathway, it is easy to envision that other microbe-host interactions are mecha-nistically linked to host disease pathogenesis, serving as the basis for the rational design of microbe-targeted therapeutics that improve human health.

## Materials and methods

**Key resources table**

| Reagent type (species) or resource | Designation | Source or reference | Identifiers | Additional information |
|---|---|---|---|---|
| Strain, strain background (*Mus musculus*) | C57BL/6J | Jackson Laboratory | Stock #: 000664; RRID:MGI:3028467 | |
| Strain, strain background (*Mus musculus*) | C57BL/6NCrl | Charles River | Stock #: 574 RRID:MGI:2159965 | |
| Genetic reagent (*Mus musculus*) | B6.Cg-Lep[ob]/J | Jackson Laboratory | Stock #: 000632; RRID:MGI:6719537 | |
| Sequence-based reagent | *Fmo3* | Sigma | | F: CCCACATGCTTTGAGAGGAG R: GGAAGAGTTGGTGAAGACCG |
| Sequence-based reagent | *Taar5* | Sigma | | F: AAAGAAAAGCTGCCAAGA R: AAGGGAAGCCAACACACA |
| Sequence-based reagent | *Arntl1* | Sigma | | F: CCAAGAAAGTATGGACACAGACAAA R: GCATTCTTGATCCTTCCTTGGT |
| Sequence-based reagent | *Nr1d1* | Sigma | | F: ATGCCAATCATGCATCAGGT R: CCCATTGCTGTTAGGTTGGT |
| Sequence-based reagent | *Cry1* | Sigma | | F: TACTGGGAAACGCTGAACCC R: ACCCCAAGCTTGTTGCCTAA |
| Sequence-based reagent | *Cry2* | Sigma | | F: GCTGGAAGCAGCCGAGGAACC R: GGGCTTTGCTCACGGAGCGA |
| Sequence-based reagent | *Per2* | Sigma | | F: GCTGACGCACACAAAGAACT R: TAGCCTTCACCTGCTTCACG |
| Sequence-based reagent | *Foxo1* | Sigma | | F: GAGCCTCCTTCAAACAGAGTAG R: GCAAATGGTAAGAAATGGCAGAG |

*Continued on next page*

*Continued*

| Reagent type (species) or resource | Designation | Source or reference | Identifiers | Additional information |
|---|---|---|---|---|
| Sequence-based reagent | *Adrb1* | Sigma | | F: TGGGAGTGTGGCTTAGTA<br>R: AATGCCTGAAGGTGCATTAAAC |
| Sequence-based reagent | *Chka* | Sigma | | F: CTGATAGCATGGCTGGGGTT<br>R: CCCAAATCAAGCTCACACACC |
| Sequence-based reagent | *Pemt* | Sigma | | F: TGTGCTGTCCAGCTTCTATG<br>R: GAAGGGAAATGTGGTCACTCT |
| Sequence-based reagent | *Pgc1a* | Sigma | | F: CCCTGCCATTGTTAAGAC<br>R: GCTGCTGTTCCTGTTTTC |
| Sequence-based reagent | *CycloA* | Sigma | | F: GCGGCAGGTCCATCTACG<br>R: GCCATCCAGCCATTCAGTC |
| Sequence-based reagent | *Gapdh* | Sigma | | F: CCTCGTCCCGTAGACAAAATG<br>R: TGAAGGGGTCGTTGATGGC |
| Commercial assay or kit | Ultra Sensitive Mouse Insulin ELISA | Crystal Chem Inc | 90080 | |
| Commercial assay or kit | Mouse Leptin ELISA | Crystal Chem Inc | 90030 | |
| Commercial assay or kit | Liver Triglycerides | Wako | 994–02891 | |
| Commercial assay or kit | Free Cholesterol | Wako | 993–02501 | |
| Commercial assay or kit | Phospholipid C | Wako | 433–26301 | |
| Commercial assay or kit | Total Cholesterol | Fisher Scientific | TR134321 | |
| Commercial assay or kit | Mouse corticosterone kit | Crystal Chem Inc | 80556 | |
| Commercial assay or kit | NEFA Kit | Wako | 999–34691, 995–34791, 991–34891, 993–35191, 276–76491 | |
| Chemical compound, drug | Iodomethylcholine | This paper | | See Materials and methods for synthesis protocol |

## Mice and experimental diets

C57Bl6/J and *Lep^ob/ob^* mice were purchased from The Jackson Laboratory (Bar Harbor, ME) and treated with a gut microbe-targeted small molecule inhibitor of choline TMA lyase called IMC (*Roberts et al., 2018*). For HFD feeding experiments, C57Bl6/J were maintained on 60% HFD alone or with the addition of 0.06% w/w IMC (D12942 and custom diet D15102401, respectively, Research Diets, Inc). For *Lep^ob/ob^* and circadian studies, mice were maintained on a standardized control chow diet (Envigo diet TD:130104) with or without supplemental with IMC (0.06% w/w) (Envigo diet # 150813). Body weight was measured weekly. Food intake was assessed by weighing the food consumed weekly divided by the number of mice in each cage and normalized to the average body weight. For studies of circadian rhythms, 9-week-old C57Bl6/J male mice were adapted for 2 weeks on a standardized minimal choline chow (Envigo diet # TD.130104) with a strict 12 hr:12 hr light:dark cycle after shipment. Mice were then maintained on standardized chow (Envigo diet # TD.130104) or the same diet supplemented with 0.06% IMC (Envigo diet # 150813). After 7 days, plasma and tissue collection were performed every 4 hs over a 24 hr period. All dark cycle necropsies were performed under red light conditions. In the obesity treatment study paradigm C57Bl6/J mice fed an HFD (Research Diets # D12942) for 6 weeks to establish obesity (body weight >35 g), and after 6 weeks of DIO mice were continued on HFD alone (Research Diets # D12942) or the same HFD-containing IMC (Research Diets #D15102401) for another 10 weeks to test whether IMC can improve obesity-related phenotypes. For studies of

the circadian patterns of gene expression and gut microbes, 9-week-old male mice were adapted for 2 weeks on chow with a strict 12 hr:12 hr light:dark cycle after shipment. Mice were randomly housed the 4 mice per cage upon arrival. Mice were then switched to standardized chow with or without 0.06% IMC by random assignment. After 7 days, plasma and tissue collection were performed every 4 hr over a 24 hr period. Sample size estimation was calculated from previous circadian quantitative PCR (qPCR) datasets from our lab. All calculations were at 95% confidence level with a power of 80% and a level of significance set at 5% (two sided), for detecting a true difference in means as stated here. The standard deviation was 0.05 units for liver, requiring a sample size of 4 to detecting a true difference between 2 means 1 and 1.1. The standard deviation of muscle was 0.35 requiring a sample size of 8 to detecting a true difference in means 1 vs. 1.5. The standard deviation in adipose was 0.38 requiring a sample size of 8 to detecting a true difference in means 1 vs. 1.5. Therefore, sample size n = 8 was used as it would be predicted to sensitive enough to detect circadian clock changes 50% of control. All mice were purchased from Jackson Labs and housed in groups of 5 per cage on corn cob bedding on ventilated racks. Before diet and drug treatments were administered, a mix of bedding was collected from all cages and then redistributed equally across all cages to set baseline microbiota diversity in all experimental mice. A minimum of two and up to four cages per condition were used for each experiment to account for cage effects. In the short-term IMC-feeding study to examine kinetic changes in the gut microbiome, 6-week-old male C57Bl/6 mice purchased from Jackson Laboratory were individually housed in cages with pooled bedding from all 16 mice enrolled in the study to normalize their baseline microbiome. Five days later, a baseline fecal sample was collected while the mice were on chow diet. After baseline collection, the mice were placed on either HFD (D12942) or HFD + 0.06% IMC (D15102401), and fecal pellets were collected every 48 hr for 6 days. For the cecal microbial transfer studies shown in Main *Figure 5*, 4-week-old germ-free B6/N mice were purchased from Charles River and placed on a sterile HFD (D12942, double-irradiated, 1.5× vitamin mix). Two weeks later, the mice were given an oral gavage of 200 mL pooled cecal contents suspended in sterile PBS + 20% glycerol from mice fed either HFD or HFD + 0.06% IMC for 6 days. A second gavage was performed 2 days later to ensure colonization. Gnotobiotic colonized mice were maintained using the Allentown Sentry SPP Cage System (Allentown, NJ) on a 14 hr:10 hr light:dark cycle. To account for potential cage effects, all data points from the kinetic and gntotobiotic 16S rRNA analysis represent mice from different cages. For all studies plasma was collected by cardiac puncture, and liver, WAT, and skeletal muscle was collected, flash-frozen, and stored at –80°C until the time of analysis. All mice harvested at the ZT14 time point underwent 24 hr cold exposure that ended 12 hr prior to necropsy. All mice were maintained in an Association for the Assessment and Accreditation of Laboratory Animal Care, International-approved animal facility, and all experimental protocols were approved by the Institutional Animal Care and use Committee of the Cleveland Clinic (Approved IACUC protocol numbers 2015–1381, 2018–1941, and 00002499).

## Synthesis of IMC iodide

IMC iodide was prepared using a previously reported method using 2-dimethylethanolamine and diiodomethane as reactants in acetonitrile followed by recrystallization from dry ethanol (*Astafev et al., 2017*). $^1$H- and $^{13}$C-NMRs of IMC were both consistent with that in the reported literature (*Astafev et al., 2017*), as well as consistent based on proton and carbon chemical shift assignments indicated below. High-resolution MS corroborated the expected cation mass and provided further evidence of structural identity.

$^1$H-NMR (600 MHz, D$_2$O): δ 5.13 (s, 2 H, -N-C$\underline{H}_2$-I), 3.90 (t, J = 4.8 Hz, 2 H, -CH$_2$-C$\underline{H}_2$-OH), 3.52 (t, J = 4.8 Hz, 2 H, -N-C$\underline{H}_2$-CH$_2$-), 3.16 (s, 6 H, -N(C$\underline{H}_3$)$_2$);
$^{13}$C-NMR (150 MHz, D$_2$O): δ 65.7 (-CH$_2$-$\underline{C}$H$_2$-OH), 55.4 (-N-$\underline{C}$H$_2$-CH$_2$-), 52.4 (-N($\underline{C}$H$_3$)$_2$), 32.3 (-N-$\underline{C}$H$_2$-I); HRMS (ESI/TOF): m/z (M$^+$) calculated for C$_5$H$_{13}$INO, 230.0036; found, 230.0033.

## Measurement of plasma TMA and TMAO and related precursors

Stable isotope dilution high-performance liquid chromatography with online tandem mass spectrometry (LC-MS/MS) was used for quantification of levels of TMAO, TMA, choline, carnitine, and *γ-butyrobetaine* in plasma, as previously described (*Hughes et al., 2010*). Their d9(methyl)-isotopologues were used as internal standards. LC-MS/MS analyses were performed on a Shimadzu 8050 triple quadrupole mass spectrometer. IMC and d2-IMC, along with other metabolites, were

monitored using multiple reaction monitoring of precursor and characteristic product ions as follows: m/z 230.0 → 58.0 for IMC; m/z 232.0 → 60.1 for d2-IMC; m/z 76.0 → 58.1 for TMAO; m/z 85.0 → 66.2 for d9-TMAO; m/z 60.2 → 44.2 for TMA; m/z 69.0 → 49.1 for d9-TMA; m/z 104.0 → 60.1 for choline; m/z 113.1 → 69.2 for d9-choline; m/z 118.0 → 58.1 for betaine; m/z 127.0 → 66.2 for d9-betaine.

## Analysis of gene expression in mouse tissues

RNA was isolated via the RNAeasy lipid tissue mini kit (Qiagen) from multiple tissues. RNA samples were checked for quality and quantity using the Bio-analyzer (Agilent). RNA-SEQ libraries were generated using the Illumina mRNA TruSEQ Directional library kit and sequenced using an Illumina HiSEQ4000 (both according to the manufacturer's instructions). RNA sequencing was performed by the University of Chicago Genomics Facility. Raw sequence files will be deposited in the Sequence Read Archive before publication (SRA). Paired-ended 1050 bp reads were trimmed with Trim Galore (v.0.3.3, http://www.bioinformatics.babraham.ac.uk/projects/trim_galore) and controlled for quality with FastQC (v.0.11.3, http://www.bioinformatics.bbsrc.ac.uk/projects/fastqc) before alignment to the *Mus musculus* genome (Mm10 using UCSC transcript annotations downloaded July 2016). Reads were aligned using the STAR alignerSTAR in single-pass mode (v.2.5.2a_modified, https://github.com/alexdobin/STAR) *Schugar, 2021* copy archived at swh:1:rev:2eb750b45549f-6b30a3a01f3b9e166e2de72a57d (*Mistry et al., 1991*) with standard parameters but specifying '–alignIntronMax 100000 –quantMode GeneCounts'. Overall alignment ranged from 88% to 99% with 61% to 75% mapping uniquely. Transcripts with fewer than one mapped read per million (MMR) in all samples were filtered out before differential expression (DE) analysis. The filtering step removed 12,692/24,411 transcripts (52%). Raw counts were loaded into R (http://www.R-project.org/) (*R Development Core Team, 2021*) and edgeR (*Wang et al., 2014*) was used to perform upper quantile, between-lane normalization, and DE analysis. Values generated with the cpm function of edgeR, including library size normalization and log2 conversion, were used in figures. Heatmaps were generated of top 50 differentially expressed transcripts using pheatmap (*Dobin et al., 2013*). Reactome-based pathway analysis was performed using an open-sourced R package: ReactomePA (*Robinson et al., 2010*). For real-time qPCR analyses ~20 mg of snap-frozen liver tissue was homogenized in the 1 mL TRIzol reagent (Thermo Fisher Scientific, Cat. No. 15596018). Furthermore, 200 mL of chloroform were added and samples were spun down at 13,000 revolutions/min for 5 min. Upper clear layer was passed through the RNeasy Mini Spin Columns (Cat. No. 74104) for clean-up of RNA according to manufacturer's instructions. DNase treatment (10 U/reaction) was performed according to the Qiagen RNeasy kit (Cat. No. 74104). Concentrations of high purity RNA were measured using Nanodrop (Thermo Fisher Scientific, ND-2000). Reverse transcription to generate cDNA was performed using qscript mastermix (Quanta- Bio Cat. No. 101414–106) as recommended by the manufacturer using 750 ng of RNA template. Resulting cDNA was diluted 10× and used in the real-time PCR using an Applied Biosystems Step One Plus thermocycler. Relative mRNA levels were calculated based on the delta-delta-CT method using the Applied Biosystems Step One Plus PCR System as we have previously described (*Kolde, 2015*; *Yu and He, 2016*; *Lord et al., 2016*; *Thomas et al., 2013*; *Schugar et al., 2017*; *Brown et al., 2010*). Primers used for qPCR are listed in the Key resources table.

## Intraperitoneal glucose tolerance testing

The mice were fasted for 4 hr before the tests. Intraperitoneal glucose tolerance testing was performed after a single IP injection of glucose (2.5 g per kg body weight for standard diet and 1 g per kg body weight for HFD and *Lep^{ob/ob}* studies). Blood glucose was then measured before (0 min) and after the injection (15, 30, 60, 120 min) using a OneTouch SelectSimple glucometer (LifeScan Inc, China).

## Measurement of plasma hormone and lipid levels

Plasma insulin (EZRMI-13K, EMD Millipore) and corticosterone (501320, Cayman Chemical) levels were measured by ELISA. Plasma non-esterified fatty acid (HR Series NEFA-HR, Wako) and triglyceride (L-Type Triglyceride M, Wako) levels were measured using enzymatic assays according to manufacturer's instructions.

## Indirect calorimetry

To measure the effects of IMC on energy expenditure and physical activity, mice were housed in metabolic cages (Oxymax CLAMS, Columbus Instruments) for indirect calorimetry measurements at room temperature (22°C). Mice were acclimated to the home cage system for 72 hr prior to data collection, and data were analyzed as previously described (*Kolde, 2015*; *Yu and He, 2016*; *Lord et al., 2016*).

## Cecal microbiome analyses

Snap-frozen cecal DNA was isolated using the MO BIO Powersoil-htp 96-well soil DNA isolation kit according to manufacturer's instructions. Region-specific primers (515F/806R) were used for amplifying the V4 region of the bacterial 16S rRNA gene for high-throughput sequencing using the Illumina HiSeq platform, paired-end 150 bp run. The reverse amplicon primer contains a 12-base Golay barcode sequence unique to each well that allows sample pooling for sequencing (*Brown et al., 2008*; *Gromovsky et al., 2018*; *Caporaso et al., 2010b*; *Caporaso et al., 2012*). More information can be found at the Earth Microbiome Project 16S rRNA Amplification Protocol where our protocols were adapted from: http://www.earthmicrobiome.org/emp-standard-protocols/16s/. Each sample was amplified in triplicate using 5 PRIME HotMaster Mix 2.5X (VWR 10847–708), combined, verified by 1.5% agarose gel, and quantitated using Pico Green dsDNA Assay Kit (Thermofisher P7589). Samples were pooled (250 ng) and cleaned using the UltraClean PCR Clean-Up Kit protocol (Mo-Bio 12500–100). The quantified amplicons were sequenced with the Illumina HiSeq 2500 at the Broad Stem Cell Research Center at the University of California – Los Angeles on two lanes. The sequences were analyzed using the open-source Python software package Quantitative Insights Into Microbial Ecology (QIIME) version 1.9.1 (*Caporaso et al., 2010b*; *Kuczynski et al., 2011*) using default parameters for each step, except where specified. Demultiplexed sequences were aligned and clustered into operational taxonomic units (OTUs) based on their sequence similarity (97% identity) using the SortMeRNA/SumaClust open reference-based OTU picking protocol in QIIME. Representative sequences for each OTU were aligned using PyNAST (a python-based implementation of NAST [*Hamady et al., 2008*] in QIIME and the Greengenes 11 database [*Kuczynski et al., 2011*]); 38,826,537 total reads were generated after removal of singleton reads and rare (<0.01% of total reads) OTUs, with an average of 776,531 reads per sample. Samples were rarefied to the depth of the sample with the lowest number of reads (86,941 sequences/sample) for beta diversity assessment only. Beta diversity was assessed using weighted UniFrac in QIIME. Adonis statistical test with 1000 permutations was used to determine the strength and statistical significance of sample groupings. LEfSe was used with default parameters on OTU tables to determine taxa that best characterize each population (*Caporaso et al., 2010a*). Significant differences in relative abundance of taxa between groups and correlations with physiological parameters were assessed using the ALDEx2 package implemented in R (*DeSantis et al., 2006*; *Segata et al., 2011*). Data were adjusted for false discovery rate (FDR) using the Benjamini-Hochberg procedure and an adjusted p-value of $p < 0.05$ was considered statistically significant. All other plots were carried out using R (r-project.org).

## Untargeted metabolomics

Mouse plasma samples were prepared for untargeted metabolomics by diluting each plasma sample 1:20 in chilled methanol containing five internal standards as listed in the table below. The samples were then centrifuged at 14,000 *g* for 20 min to precipitate out the protein pellet. The supernatant was recovered and subjected to LC-MS analysis. One-microliter aliquots taken from each sample were pooled and this QC standard was analyzed every 10th injection. The untargeted metabolomics was performed by injecting 7 µL of each sample onto a 10 cm C18 column (Thermo Fisher CA) coupled to a Vanquish UHPLC running at 0.25 mL/min using water and 0.1% formic acid as solvent A and acetonitrile and 0.1% formic acid as solvent B. The 15 min gradient used is given below. The Orbitrap Q Exactive HF was operated in positive and negative electrospray ionization modes in different LC-MS runs over a mass range of 50–750 Da using full MS at 120,000 resolution. Data-dependent acquisitions were obtained on the pooled QC sample. The DDA acquisition (DDA) include MS full scans at a resolution of 120,000 and HCD MS/MS scans taken on the top 10 most abundant ions at a resolution of 30,000 with dynamic exclusion of 4.0 s and the apex trigger set at 2.0–4.0 s. The resolution of the MS2 scans were taken at a stepped NCE energy of 20.0, 30.0, and 45.0. XCMS was used to deconvolute the data using 5 ppm consecutive scan error, 5–60 s as minimum and maximum peak width, S/N

threshold of 10, and span of 0.2 in positive mode and span of 0.4 in negative mode for retention time correction. The resulting peak table was further analyzed via MetaboLyzer (*Mak et al., 2015*). Briefly, the ion presence threshold was set at 0.7 in each study group. Data were then log-transformed and analyzed for statistical significance via non-parametric Mann-Whitney U-test (FDR-corrected p-value < 0.05). Ions present in just a subset of samples were analyzed as categorical variables for presence status via Fisher's exact test. All p-values were corrected via the Benjamini-Hochberg step-up procedure for FDR correction. The data was then utilized for PCA, putative identification assignment, and pathway enrichment analysis via KEGG. In this dataset 7665 spectral features were detected, from which 1151 features were putatively assigned an identification in HMDB within a pre-defined 7 ppm m/z error window. Also, the MS/MS spectra of 120 of these features matched with a score of greater than 50–120% unique compounds on the *mzCloud* database. Given the complicated nature of comparing the global metabolome across two treatment groups and six circadian time points, we used an algorithm called SPICA (*Fernandes et al., 2013*), to reveal subtle differences the plasma metabolome kinetically. Pairwise analysis was conducted between all adjacent time points (ZT2, ZT6, ZT10, ZT14, ZT18, and ZT22) for each treatment group (chow control vs. chow + IMC), resulting in a total of 10 comparisons made (five for chow and five for chow + IMC). Differences in the global plasma metabolome for each pairwise comparison were quantified via receiver operating characteristic (ROC) curve construction and area under the curve (AUC) calculations via Monte Carlo cross validation procedures in SPICA (*Fernandes et al., 2013*). This analysis revealed that while all adjacent time points were roughly equally differentiated in the chow control data with an AUC averaging 0.884, this was not the case in the IMC-treated group. The AUC calculated for the IMC-treated group when comparing T10 vs. T14 was much greater than that of the chow-fed control group at the same time points (0.964 vs. 0.869), meaning the differences in the plasma metabolome between these two time points were much more pronounced in the IMC-treated mice compared to chow controls. Furthermore, The AUCs calculated for the subsequent two time point comparisons (T14 vs. T18 and T18 vs. T22) were much lower in the IMC-treated group (0.624 and 0.615, respectively) when compared to the chow control group (0.842 and 0.899, respectively), implying that the T14, T18, and T22 time points were poorly differentiated by IMC treatment. As a result, subsequent data analyses focused on this T10 to T14 transition, which also coincided with when the mice began to eat. The statistically significant positively charged and negatively charged spectral features which were present in more than 70% of the samples at ZT14 are shown in red in the volcano plots of *Figure 4*. These features were then putatively identified in the Human Metabolome and the KEGG databases using their accurate mass-to-charge (m/z) values within a 7 ppm error window. The KEGG annotated pathways associated with these putative metabolites were then identified. *Figure 4c* represents such KEGG pathways associated with the negatively charged spectral features in this study. This figure displays the KEGG metabolic pathways with the highest statistical significance to which the ions were assigned. The blue and red bars are the unadjusted and the FDR-adjusted −log of p-values, respectively, while the orange line marks the significance threshold. This figure shows that lipid metabolism is the most perturbed metabolic pathway at ZT14.

## Data analyses for circadian rhymicity (cosinor analyses)

A single cosinor analysis was performed as previously described (*Fernandes et al., 2014*; *Cornelissen, 2014*). Briefly, a cosinor analysis was performed on each sample using the equation for cosinor fit as follows:

$$Y(t) = M + A\cos\left(2\theta/\tau + \phi\right)$$

where M is the MESOR (midline statistic of rhythm, a rhythm adjusted mean), A is the amplitude (a measure of half the extent of the variation within the cycle), $\Phi$ is the acrophase (a measure of the time of overall highest value), and $\tau$ is the period. The fit of the model was determined by the residuals of the fitted wave. After a single cosinor fit for all samples, linearized parameters were then averaged across all samples allowing for calculation of delinearized parameters for the population mean. A 24 hr period was used for all analysis. Comparison of population MESOR, amplitude, and acrophase was performed as previously described (*Zhu et al., 2017*). Comparisons are based on F-ratios with degrees of freedom representing the number of populations and total number of subjects.

All analyses were done in R v.4.0.2 using the cosinor and cosinor2 packages (*Nelson et al., 1979*; *Bingham et al., 1982*; *Sachs et al., 2013*, *Mutak, 2018*, *Sachs, 2014*).

## Statistical analysis

All data were analyzed using either one-way or two-way analysis of variance (ANOVA) where appropriate, followed by either a Tukey's or Student's t-tests for post hoc analysis. Differences were considered significant at $p < 0.05$. All mouse data analyses were performed using GraphPad Prism 6 (La Jolla, CA) software.

## Acknowledgements

This work was supported by National Institutes of Health grants R01 HL120679 (JMB), P01 HL147823 (JMB, SLH), P50 AA024333 (JMB), U01 AA026938 (JMB), R01 DK130227 (JMB), P50 CA150964 (JMB), R01 HL103866 (SLH), R01 HL147883 (AJL), R01 HL144651 (AJL and ZW), R01 HL130819 (ZW), F32 DK122623 (CMG), T32 DK007307 (CMG), a Leducq Transatlantic Networks of Excellence Award (SLH), and the American Heart Association (Postdoctoral Fellowships 17POST3285000 to RNH and 15POST2535000 to RCS). Development of some of the mass spectrometry methods reported here was supported by generous pilot grants from the Clinical and Translational Science Collaborative of Cleveland (4UL1TR000439) from the National Center for Advancing Translational Sciences (NCATS) component of NIH and the NIH Roadmap for Medical Research, the Case Comprehensive Cancer Center (P30 CA043703), the VeloSano Foundation, and a Cleveland Clinic Research Center of Excellence Award. The authors would like to thank Ken Kula from the Department of Medical Art and Photography at Cleveland Clinic for the illustrations within this manuscript.

## Additional information

### Competing interests

Jennifer A Buffa: reports being eligible to receive royalty payments for inventions or discoveries related to cardiovascular therapeutics from the Proctor & Gamble Co. Jose Carlos Garcia-Garcia: Employee of Procter & Gamble Company. Zeneng Wang, Stanley L Hazen: reports being named as co-inventor on pending and issued patents 20200121615 held by the Cleveland Clinic relating to cardiovascular diagnostics and therapeutics. Reports being a paid consultant for Procter & Gamble, having received research funds from Procter & Gamble, Roche Diagnostics, and being eligible to receive royalty payments for inventions or discoveries related to cardiovascular diagnostics or therapeutics from Cleveland Heart Lab and Procter & Gamble. The other authors declare that no competing interests exist.

### Funding

| Funder | Grant reference number | Author |
|---|---|---|
| National Institute of Diabetes and Digestive and Kidney Diseases | R01 DK120679 | Jonathan Mark Brown |
| National Heart, Lung, and Blood Institute | P01 HL146823 | Stanley L Hazen |
| National Institute on Alcohol Abuse and Alcoholism | P50 AA024333 | Jonathan Mark Brown |
| National Institute on Alcohol Abuse and Alcoholism | U01 AA026938 | Jonathan Mark Brown |
| National Institute of Diabetes and Digestive and Kidney Diseases | R01 DK130227 | Jonathan Mark Brown |
| National Cancer Institute | P50 CA150964 | Jonathan Mark Brown |

| Funder | Grant reference number | Author |
| --- | --- | --- |
| National Heart, Lung, and Blood Institute | R01 HL103866 | Stanley L Hazen |
| National Heart, Lung, and Blood Institute | R01 HL147883 | Aldons J Lusis |
| National Heart, Lung, and Blood Institute | R01 HL144651 | Aldons J Lusis |
| National Heart, Lung, and Blood Institute | R01 HL130819 | Zeneng Wang |
| National Institute of Diabetes and Digestive and Kidney Diseases | F32 DK122623 | Christy M Gliniak |
| National Institute of Diabetes and Digestive and Kidney Diseases | T32 DK007307 | Christy M Gliniak |
| Leducq Transatlantic Network of Excellence award | No grant number | Stanley L Hazen |
| American Heart Association | 17POST3285000 | Robert N Helsley |
| American Heart Association | 15POST2535000 | Rebecca C Schugar |
| Clinical and Translational Science Collaborative of Cleveland, School of Medicine, Case Western Reserve University | 4UL1TR000439 | Belinda Willard |
| Case Comprehensive Cancer Center, Case Western Reserve University | P30 CA043703 | Jonathan Mark Brown |
| National Institutes of Health | R01 HL120679 | Jonathan Mark Brown |
| National Institutes of Health | P01 HL147823 | Jonathan Mark Brown Stanley L Hazen |

The funders had no role in study design, data collection and interpretation, or the decision to submit the work for publication.

## Author contributions

Rebecca C Schugar, Conceptualization, Data curation, Formal analysis, Funding acquisition, Investigation, Methodology, Project administration, Resources, Software, Supervision, Validation, Visualization, Writing – review and editing; Christy M Gliniak, Conceptualization, Data curation, Formal analysis, Investigation, Methodology, Writing - original draft, Writing – review and editing; Lucas J Osborn, Conceptualization, Data curation, Formal analysis, Investigation, Methodology, Writing – review and editing; William Massey, Robert N Helsley, Amanda L Brown, Chase Neumann, Amy McMillan, Jennifer A Buffa, Margarete Mehrabian, Maryam Goudarzi, Belinda Willard, Tytus D Mak, Andrew R Armstrong, Garth Swanson, Data curation, Investigation, Methodology, Writing – review and editing; Naseer Sangwan, Anthony Horak, Rakhee Banerjee, Danny Orabi, Amy Burrows, Chelsea Finney, Kevin K Fung, Frederick M Allen, Daniel Ferguson, Anthony D Gromovsky, Kendall Cook, James T Anderson, Investigation, Methodology, Writing – review and editing; Ali Keshavarzian, Jose Carlos Garcia-Garcia, Zeneng Wang, Conceptualization, Data curation, Investigation, Methodology, Writing – review and editing; Aldons J Lusis, Data curation, Formal analysis, Investigation, Methodology, Writing – review and editing; Stanley L Hazen, Conceptualization, Data curation, Formal analysis, Methodology, Project administration, Resources, Supervision, Writing – review and editing; Jonathan Mark Brown, Conceptualization, Data curation, Formal analysis, Funding acquisition, Investigation, Methodology, Project administration, Resources, Software, Supervision, Validation, Visualization, Writing - original draft, Writing – review and editing

## Author ORCIDs

Rebecca C Schugar http://orcid.org/0000-0001-5908-2213
Christy M Gliniak http://orcid.org/0000-0003-3806-6112
Lucas J Osborn http://orcid.org/0000-0003-0077-9192
William Massey http://orcid.org/0000-0002-2087-6048
Robert N Helsley http://orcid.org/0000-0001-5000-3187
Ali Keshavarzian http://orcid.org/0000-0002-7969-3369
Jonathan Mark Brown http://orcid.org/0000-0003-2708-7487

## Ethics

All mice were maintained in an Association for the Assessment and Accreditation of Laboratory Animal Care, International-approved animal facility, and all experimental protocols were approved by the Institutional Animal Care and use Committee of the Cleveland Clinic. (Approved IACUC protocol numbers 2015-1381, 2018-1941, and 00002499).

## Decision letter and Author response

Decision letter https://doi.org/10.7554/eLife.63998.sa1
Author response https://doi.org/10.7554/eLife.63998.sa2

## Additional files

### Supplementary files
• Transparent reporting form

### Data availability

RNA sequencing data has been deposited in GEO under accession code GSE157925. Microbiome data were submitted to the European Nucleotide Archive under accession code PRJEB48232.

The following datasets were generated:

| Author(s) | Year | Dataset title | Dataset URL | Database and Identifier |
|---|---|---|---|---|
| Brown JM, Schugar R, Gliniak C, Neumann C | 2021 | Gut Microbe-Targeted Choline Trimethylamine Lyase Inhibitors Improves Obesity Via Rewiring of Host Circadian Rhythms | https://www.ncbi.nlm.nih.gov/geo/query/acc.cgi?acc=GSE157925 | NCBI Gene Expression Omnibus, GSE157925 |
| Schugar RC, Gliniak CM, Osborn LJ, Massey W, Sangwan N, Horak A, Banerjee R, Orabi D, Helsley RN, Brown AL, Burrows A, Finney C, Fung KK, Allen FM, Ferguson D, Gromovsky AD, Neumann C, Cook K, McMillan A, Buffa JA, Anderson JT, Mehrabian M, Goudarzi M, Willard B, Mak TD, Armstrong AR, Swanson G, Keshavarzian A, Garcia-Garcia JC, Wang Z, Lusis AJ, Hazen SL, Brown JM | 2021 | Gut Microbe-Targeted Choline Trimethylamine Lyase Inhibition Improves Obesity Via Rewiring of Host Circadian Rhythms | https://www.ebi.ac.uk/ena/browser/view/PRJEB48232?show=reads | European Nucleotide Archive, ERP132574 |

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
