## [Editor Report]

Schugar et al. present data on the effects of a small molecule inhibitor (iodomethylcholine, IMC) of bacterial choline metabolism. This extends prior work from this team of scientists to focus on obesity-related phenotypes. They report a decrease in body weight in a diet-induced obesity model accompanied by lower insulin and improved glucose control. Remarkably, they also observed phenotypes in the ob/ob (leptin-deficient) model, which has severe obesity. They go on to describe additional phenotypes in IMC treated and control mice: gut microbiota, gene expression, and metabolomics, with a focus on circadian rhythm. Taken together, these data support the potential therapeutic value of IMC for treating obesity and associated metabolic diseases.

---

## [Decision Letter]

**Decision letter after peer review:**

Thank you for submitting your article "Gut Microbe-Targeted Choline Trimethylamine Lyase Inhibition Improves Obesity Via Rewiring of Host Circadian Rhythms" for consideration by *eLife*. Your article has been reviewed by 3 peer reviewers, one of whom is a member of our Board of Reviewing Editors, and the evaluation has been overseen by Wendy Garrett as the Senior Editor. The reviewers have opted to remain anonymous.

The reviewers have discussed the reviews with one another and the Reviewing Editor has drafted this decision to help you prepare a revised submission.

Summary:

Schugar et al. present data on the impact of a small molecule inhibitor (iodomethylcholine, IMC) of bacterial choline metabolism. This extends prior work from this team of scientists to focus on obesity-related phenotypes. They report a decrease in body weight in a diet-induced obesity model accompanied by lower insulin and improved glucose control. Remarkably, they also observed phenotypes in the ob/ob (leptin-deficient) model, which has severe obesity. They go on to describe additional phenotypes in IMC treated and control mice: gut microbiota, gene expression, and metabolomics, with a focus on circadian rhythm. Taken together, these data support the potential therapeutic value of IMC for treating obesity and associated metabolic diseases.

Essential revisions:

While each of these datasets is intriguing, they haven't been integrated in such a way to explain the underlying mechanisms responsive for the initial phenotypes. As such, the extent to which these data change our understanding of obesity and associated metabolic diseases remains unclear.

Given the diverse datasets and concepts in the current manuscript, it would not be feasible to reduce each component to its mechanistic basis in a single paper. Instead, we propose that the authors choose one of the following options:

Option 1. Causal data that IMC effects on obesity are gut microbiome dependent.

It remains unclear if the phenotypes are related at all to CutC activity or a downstream consequence of the observed differences in the gut microbiota. There is also no explanation provided as to why IMC would impact the gut microbiota so much. Is choline metabolism really that important or is this potentially indicative of off-target effects? These questions could be addressed using gnotobiotic mice +/- CutC encoding strains in combination with pharmacological manipulation of choline or its downstream metabolites.

There is also no causal data presented to suggest that the IMC-associated gut microbiota impacts obesity. This could be readily done using microbiota transplantations to germ-free mice and/or through re-colonizing IMC treated mice with bacteria that are at decreased abundance following treatment. IMC treatment of germ-free or antibiotic-depleted mice would also help to test if there are direct effects of IMC on the host that impact body weight and adiposity; for example, decreasing food intake due to altered taste.

Option 2. Causal data that IMC effects are dependent on circadian networks.

The authors observe that a number of different genes are altered in a tissue-dependent manner upon IMC exposure, including shifts in Bmal1 and Rev-erba. Tissue-specific Bmal1 and global Rev-erba mice are available and could be used to determine the mechanistic basis of the interaction between the core clock machinery, TMA oscillations from gut microbes, as well as resulting downstream phospholipid species. Do tissue specific Bmal1 (Arntl) or Reverba knock-outs exhibit unique responsiveness to microbial CutC activity and TMA to TMAO levels/conversions at baseline or following HF-diet feeding?

Review Gréchez-Cassiau A, Feillet C, Guérin S, Delaunay F. The hepatic circadian clock regulates the choline kinase α gene through the BMAL1-REV-ERBα axis. (Chronobiol Int. 2015;32(6):774-84. doi: 10.3109/07420528.2015.1046601. Epub 2015 Jun 30. PMID: 26125130).

[Editors' note: further revisions were suggested prior to acceptance, as described below.]

Thank you for submitting your article "Gut Microbe-Targeted Choline Trimethylamine Lyase Inhibition Improves Obesity Via Rewiring of Host Circadian Rhythms" for consideration by *eLife*. Your article has been reviewed by two peer reviewers, and the evaluation has been overseen by a Reviewing Editor and Wendy Garrett as the Senior Editor. The following individual involved in review of your submission has agreed to reveal their identity: Sudha Biddinger (Reviewer #2).

Essential revisions:

See the itemized comments from reviewer #3 for additional points to clarify in your resubmitted paper. Please note that we do not require any additional data or experiments at this time.

*Reviewer #3:*

Schugar et al., have presented studies examining how a small molecule (iodomethylcholine, IMC) prevents microbially-mediated TMA production via CutC inhibition to protect against diet-induced obesity. The authors use a variety of tools to interrogate the mechanistic underpinnings of IMC's impact on host metabolism, and suggest this is in part mediated by IMC-induced shifts in gut microbes that impact circadian outputs. The authors have generated additional data to address the initial set of reviewers' comments. In many cases, the manuscript has been strengthened. However, several points still remain regarding the connection between circadian aspects of the studies surrounding the capacity of IMC to shift circadian rhythms and subsequent metabolic outputs. For instance, many of the studies examining IMC's ability to shift circadian rhythms were performed in chow-fed animals. The meaning of shifts in circadian rhythms in regular chow-fed animals as they pertain to metabolic outputs in lean animals remains puzzling. Identical studies performed in HF-fed mice would clarify several key conclusions the authors are attempting to draw from these data.

1) Can the authors discuss what the implications are of IMC-induced shifts in circadian gene expression in regular chow-fed mice relative to regular chow untreated counterparts? Given that chow-fed mice aren't obese, the interpretation is a bit unclear regarding the protective metabolic impacts. This is also the case regarding phospholipid profiles presented in Figure 4 – what is the meaning of a 12-hour phase-shift in choline containing phospholipids in already lean animals?

2) Related to the comment #1, Figure 1—figure supplement 4 contains data where animals were fed HF diet followed by treatment w/ or w/o IMC to test therapeutic capacity – can the authors discuss the circadian aspects of gene expression? Are the phase and amplitude shifts induced by IMC also evident in HF-fed animals?

3) Despite the response to Reviewers' comments regarding the focus on adipose tissue vs. liver, the focus and findings in muscle are intriguing. Could the authors discuss and quantify the 12-hour phase shift in muscle Taar5 expression? Again, given that chow-fed animals are not obese, it is unclear what the gene expression changes indicate in already lean animals and it would be ideal to show under HF feeding conditions that this change in rhythmicity persists.

4) Line 279 – 282: The authors make a claim about microbial oscillations, however, the resolution of sample collections every 48 hours is not refined enough to make statements regarding microbial rhythms. It is possible the authors misinterpreted my initial comment regarding the need for 48-hour fecal collections (48-hour fecal collections refer to fecal collections from the same animal every 4 to 6 hours over 2 consecutive LD cycles, rather than fecal samples collected every 48 hours). Further, simply examining ZT2 vs. ZT14 does not provide enough sampling time points to identify rhythms. To truly examine microbial oscillations, resolution needs to be refined to every 4 hours over a 24 hour period at a minimum for cecal contents and every 4 to 6 hours over a 48 hour period for stool. These more resolved data are shown in a high-resolution figure at the end of the manuscript proof on page 81 and is presumably meant to be Figure 2—figure supplement 2 (although the high-resolution images are not labeled with figure #). This high-resolution figure seems to be missing from the figures shown beginning on page 55 of the proof. Can the authors resolve this discrepancy regarding this supplemental figure?

5) Similar to comment 4, it would be ideal to show that the microbial rhythmicity changes induced by IMC treatment also persist in HF-fed animals and are not just observed under chow-fed conditions.

6) In reading through the methods, germ-free mice used for conventionalization were maintained under a different light schedule (14:10LD) relative to the conventionally-raised mice (12:12LD). Yet, all animals were harvested at similar timepoints. Can the authors at a minimum, mention this in the discussion as a potential caveat in the interpretation of the data? Further, the authors are drawing broad claims regarding circadian rhythmicity of microbiota as well as host parameters based on two timepoints in the conventionalized study, which is not acceptable. A minimum of 3 timepoints (at least to make the initial claim) are needed to draw conclusions regarding rhythms in gut microbes and host parameters. Suggest softening claims regarding these data as they relate to circadian dynamics.

7) Regarding the conventionalization study – can the authors provide statistics for the engraftment efficiency relative to the donors? The stacked bars from the donors are shown in panel f in Figure 2—figure supplement 1, but it does not seem like all animals colonized to an equal extent (i.e., conventionalized mice at ZT2 from the HFD CCTx group shown in Figure 5, panel b appear to be missing Blautia). For the mice that were collected at ZT2 vs. ZT14, were all the animals harvested from a single cage? Or were they spread out across cages? This is important for conventionalization, since colonization efficiency can be unique from cage to cage, creating artifacts within a timepoint if not accounted for.

---

## [Author Response]

Essential revisions:While each of these datasets is intriguing, they haven't been integrated in such a way to explain the underlying mechanisms responsive for the initial phenotypes. As such, the extent to which these data change our understanding of obesity and associated metabolic diseases remains unclear.Given the diverse datasets and concepts in the current manuscript, it would not be feasible to reduce each component to its mechanistic basis in a single paper. Instead, we propose that the authors choose one of the following options:

We sincerely appreciate the reasonable nature of this request. We agree both options below are critically important questions. To make this revision feasible with one year, we chose option 1 below, and feel the results have strengthened the manuscripts conclusions.

Option 1. Causal data that IMC effects on obesity are gut microbiome dependent.It remains unclear if the phenotypes are related at all to CutC activity or a downstream consequence of the observed differences in the gut microbiota. There is also no explanation provided as to why IMC would impact the gut microbiota so much. Is choline metabolism really that important or is this potentially indicative of off-target effects? These questions could be addressed using gnotobiotic mice +/- CutC encoding strains in combination with pharmacological manipulation of choline or its downstream metabolites.There is also no causal data presented to suggest that the IMC-associated gut microbiota impacts obesity. This could be readily done using microbiota transplantations to germ-free mice and/or through re-colonizing IMC treated mice with bacteria that are at decreased abundance following treatment. IMC treatment of germ-free or antibiotic-depleted mice would also help to test if there are direct effects of IMC on the host that impact body weight and adiposity; for example, decreasing food intake due to altered taste.

We thank the reviewer for bringing up this point. We agree. To address the concerns raised in Option 1 above, we performed microbiota transplantation studies where we transferred control or IMC-treated mouse cecal contents into naïve germ-free mice and studied the downstream consequences on high fat diet-induced obesity and circadian disruption in re-colonized mice.

The results of this new study are included in main Figure 5 in the revised manuscript. Transplantion of cecal contents from mice previously treated with the TMA lyase inhibitor IMC was able to transmit reduced adiposity in recipient mice. Furthermore, cecal transplant from IMC-treated mice was also able to reorganize core circadian gene expression in the skeletal muscle of recipient mice. Collectively, our studies demonstrate for the first time that gut microbial TMA(O) production impacts core circadian metabolic pathways and obesity susceptibility in the host. Moreover, these new studies reveal the beneficial effects observed on adiposity and circadian rhythm alterations in the host following inhibition of gut microbial TMA production (i.e. with IMC) can be attributed, in part, to the microbiome altering properties of the drug.

Option 2. Causal data that IMC effects are dependent on circadian networks.The authors observe that a number of different genes are altered in a tissue-dependent manner upon IMC exposure, including shifts in Bmal1 and Rev-erba. Tissue-specific Bmal1 and global Rev-erba mice are available and could be used to determine the mechanistic basis of the interaction between the core clock machinery, TMA oscillations from gut microbes, as well as resulting downstream phospholipid species. Do tissue specific Bmal1 (Arntl) or Reverba knock-outs exhibit unique responsiveness to microbial CutC activity and TMA to TMAO levels/conversions at baseline or following HF-diet feeding?Review Gréchez-Cassiau A, Feillet C, Guérin S, Delaunay F. The hepatic circadian clock regulates the choline kinase α gene through the BMAL1-REV-ERBα axis. (Chronobiol Int. 2015;32(6):774-84. doi: 10.3109/07420528.2015.1046601. Epub 2015 Jun 30. PMID: 26125130).

We agree these are obvious next steps with this work. However, given the constraints of the COVID-19 pandemic, including restrictions on mouse housing, we have elected “option 1”. These follow up studies are planned in 2022 and beyond, and will allow us to further delineate mechanisms by which gut microbial TMA production may influence the host circadian clock and circadian-dependent regulation of lipid metabolism.

[Editors' note: further revisions were suggested prior to acceptance, as described below.]

Reviewer #3:Schugar et al., have presented studies examining how a small molecule (iodomethylcholine, IMC) prevents microbially-mediated TMA production via CutC inhibition to protect against diet-induced obesity. The authors use a variety of tools to interrogate the mechanistic underpinnings of IMC's impact on host metabolism, and suggest this is in part mediated by IMC-induced shifts in gut microbes that impact circadian outputs. The authors have generated additional data to address the initial set of reviewers' comments. In many cases, the manuscript has been strengthened. However, several points still remain regarding the connection between circadian aspects of the studies surrounding the capacity of IMC to shift circadian rhythms and subsequent metabolic outputs. For instance, many of the studies examining IMC's ability to shift circadian rhythms were performed in chow-fed animals. The meaning of shifts in circadian rhythms in regular chow-fed animals as they pertain to metabolic outputs in lean animals remains puzzling. Identical studies performed in HF-fed mice would clarify several key conclusions the authors are attempting to draw from these data.

We completely agree with this reviewer concern, and in fact have performed studies with IMC both in chow-fed mice as well as in mice fed matched low-fat and high-fat diets. Although the majority of circadian data presented are in chow-fed mice, we include new data in this revision in the high fat-fed cohorts. These data confirm that IMC alters circadian rhythms in metabolism in both chow and high fat-fed mice, and we importantly now show that the ability of IMC to alter host circadian rhythms can be transferred to germ-free mice implicating the gut microbiome in the drug’s anti-obesity effects.

1) Can the authors discuss what the implications are of IMC-induced shifts in circadian gene expression in regular chow-fed mice relative to regular chow untreated counterparts? Given that chow-fed mice aren't obese, the interpretation is a bit unclear regarding the protective metabolic impacts.

We thank the reviewer for bringing up this important point. We performed extensive circadian analysis in a chow-fed cohort simply due to the fact that >99% of all published studies in the literature use rodent chow as a base diet. We feel it is important to include these data in this initial description of IMC circadian links, but we also include new data in this revision showing the IMC can alter circadian oscillations in metabolism in high fat-fed mice (Figure 3—figure supplement 2).

This is also the case regarding phospholipid profiles presented in Figure 4 – what is the meaning of a 12-hour phase-shift in choline containing phospholipids in already lean animals?

Due to the complexity of the interorgan crosstalk (gut microbiome, liver, adipose, etc.) in IMC-treated mice, we hesitate to overinterpret the metabolic impact of altered oscillation in phosphatidylcholine (PC) metabolism in this manuscript. In fact, many follow up studies will be required to understand how TMA lyase inhibitors impact PC metabolism throughout the body. Although beyond the scope of this manuscript, we are actively investigating how the gut microbial TMAO pathway can alter hepatic phosphatidylcholine (PC) homeostasis. PC plays many key roles in the liver and systemic distribution of lipids, including its key role in hepatic biliary lipid secretion, very low density lipoprotein (VLDL) secretion, and PC is the major structural lipid in all mammalian cell membranes so can have key roles in lipid signaling and cell function. Therefore in follow up studies, we plan to examine the effect of exogenous TMA and/or TMA lyase drug treatment of biliary lipid section, VLDL production, and cell signaling. However, these are well beyond the scope of this current manuscript. The key new finding from this current work is that TMA lyase inhibition alters the normal circadian oscillations in circulating lysophosphatidylcholine (LPC) and PC, as well as circadian oscillation in the hepatic gene expression of key PC biosynthetic enzymes (choline kinase a, CKa and phosphatidylethanolamine methyltransferase, PEMT). It is important to note that we arose at this conclusion using unbiased LC-MS/MS-based metabolomic approaches (i.e. followed unbiased data to our conclusions), and we feel strongly that the PC altering aspect of TMA lyase inhibitors likely play an important role in shaping systemic lipid metabolism. Although we did not do an extensive dive in PC metabolism in high fat diet-fed mice here, we did find that the circadian expression of another PC metabolizing enzyme betaine homocysteine methyltransferase (BHMT) was altered in IMC-treated mice. These new data are included in the revised manuscript in Figure 3—figure supplement 2. We have now embarked on follow up studies examining the potential for bacterially-derived TMA to alter circadian rhythms in hepatic PC metabolism via the host GPCR trace amine-associated 5 (TAAR5).

2) Related to the comment #1, Figure 1—figure supplement 4 contains data where animals were fed HF diet followed by treatment w/ or w/o IMC to test therapeutic capacity – can the authors discuss the circadian aspects of gene expression? Are the phase and amplitude shifts induced by IMC also evident in HF-fed animals?

We completely agree with the reviewer’s interest about results from HFD cohorts. We also understand it is relevant for the reviewer to mention HFD studies in several contexts pertaining to this manuscript. Unfortunately, the samples collected for Figure 1 —figure supplement 4 were performed at a single timepoint and therefore cannot be used to address the question above. However, we have provided new data in the revised manuscript showing that IMC can alter circadian fluctuations in circulating metabolic hormones, as well as key metabolic genes in the liver and skeletal muscle (Figure 3—figure supplement 2).

3) Despite the response to Reviewers' comments regarding the focus on adipose tissue vs. liver, the focus and findings in muscle are intriguing. Could the authors discuss and quantify the 12-hour phase shift in muscle Taar5 expression? Again, given that chow-fed animals are not obese, it is unclear what the gene expression changes indicate in already lean animals and it would be ideal to show under HF feeding conditions that this change in rhythmicity persists.

We agree that circadian expression of *Taar5* in muscle was an interesting finding, and worth further exploration. We performed cosinor analysis for skeletal muscle *Taar5* gene expression (Figure 3—figure supplement 1 – Panel D). Acrophase shifted from about 20.4 hours in control mice to 0 in mice fed IMC. In the discussion we describe that it is tempting to speculate that the oscillation of muscle Taar5 may be sensitive to the oscillation of serum TMA levels, given that IMC inhibits the levels and oscillation of serum TMA. Also, as described above we have included new data in high fat fed mice, showing that IMC treatment can reorganize plasma hormone levels and circadian gene expression in skeletal muscle (Figure 3—figure supplement 2).

4) Line 279 – 282: The authors make a claim about microbial oscillations, however, the resolution of sample collections every 48 hours is not refined enough to make statements regarding microbial rhythms. It is possible the authors misinterpreted my initial comment regarding the need for 48-hour fecal collections (48-hour fecal collections refer to fecal collections from the same animal every 4 to 6 hours over 2 consecutive LD cycles, rather than fecal samples collected every 48 hours).

Indeed we did misinterpret the original reviewer concern to mean that we should collect feces every 48 hours and perform 16S microbiome analysis. The original comment was “Could the authors perform 48-hour fecal collections – (1) before, (2) early following IMC administration onset, and (3) late following IMC administration onset to show if gut microbiota oscillations observed at baseline pre-administration the same or unique post-administration and before overt metabolic and physiological changes are occurring?” We interpreted to mean “perform 48-hour fecal collections” immediately after the drug was given. Therefore, as requested we performed fecal collections after drug exposure for 4 48-hour time periods. This was not a trivial experiment and quite costly to perform longitudinal 16S sequencing. The resulting data clearly show that drug effect is seen quite rapidly and maintain throughout the time period assayed, which we feel help strengthen some of the microbiome-related conclusions of this work. On top of this experiment performed to be responsive to reviewer concerns, we performed another experiment where cecal microbiome analysis was performed every 4 hours over a 24 hour time course to examine more kinetic changes in the gut microbiome in the same mice (Figure 2—figure supplement 2). Although the mice have been exposed to drug for a week prior to cecum collection, these data clearly show oscillations in gut microbial populations that agrees with other published results, and show that IMC treatment significantly alters the normal circadian oscillations in gut microbial communities. To avoid confusion, we have removed any unintentional claims we may have made concerning microbial oscillation, unless we collected data every 4 hours over a 24-hour period, taking great care not to claim oscillation changes occurred for data measured only at one zeitgeber (ZT) time points.

Further, simply examining ZT2 vs. ZT14 does not provide enough sampling time points to identify rhythms. To truly examine microbial oscillations, resolution needs to be refined to every 4 hours over a 24 hour period at a minimum for cecal contents and every 4 to 6 hours over a 48 hour period for stool. These more resolved data are shown in a high-resolution figure at the end of the manuscript proof on page 81 and is presumably meant to be Figure 2—figure supplement 2 (although the high-resolution images are not labeled with figure #). This high-resolution figure seems to be missing from the figures shown beginning on page 55 of the proof. Can the authors resolve this discrepancy regarding this supplemental figure?

We apologize for confusion, but the experiment we did is exactly what the reviewer requests. The Reviewer suggests “resolution needs to be refined to every 4 hours over a 24 hour period at a minimum for cecal contents.…” In the experiment shown in Figure 2—figure supplement 2 we did exactly this, collecting cecal contents from experimental mice every 4 hours over a 24 hour period. This allowed us to utilize JTK_cycle to analyze the rhythmic oscillation within the gut microbiome.

5) Similar to comment 4, it would be ideal to show that the microbial rhythmicity changes induced by IMC treatment also persist in HF-fed animals and are not just observed under chow-fed conditions.

We unfortunately did not perform 16S analyses in the high fat diet-fed cohorts treated with IMC, and respectfully suggest that this is beyond the scope of this current manuscript. After last revision the editorial summary stated “Please note that we do not require any additional data or experiments at this time.” We feel strongly that even if these costly data were generated and included they will not dramatically alter any of the interpretations made. It is important to note that any additional 16S or metagenomic data generated would only allow for association – not causation.

6) In reading through the methods, germ-free mice used for conventionalization were maintained under a different light schedule (14:10LD) relative to the conventionally-raised mice (12:12LD). Yet, all animals were harvested at similar timepoints. Can the authors at a minimum, mention this in the discussion as a potential caveat in the interpretation of the data?

We agree that the light cycle is extremely important to consider designing any rodent experiment. The reason for this light cycle difference was due to institutional standard operating procedure. At the Cleveland Clinic, our mouse gnotobiotic facility is maintained on a 14:10 light cycle to maintain a diverse array of germ-free mouse lines. Therefore, we were not afforded the ability to switch the light:dark cycle in an area that serves many different laboratories. Given previous reviewer concerns to establish a gut microbial link to the observed IMC-induced metabolic reorganization, we felt it was imperative to perform a microbiome transfer experiment in germ-free mice under the most rigorous conditions allowed at our institution. The 2-hour light cycle difference is not a particular concern in our case, as we do not compare phase shifts between the germ-free and conventional mice. For us the consistency of the cycle was the critical variable, and in fact find similar alterations in circadian oscillation in host metabolism. We continued to compare ZT2 versus ZT14 in the germ-free mice, because although there were clear IMC-induced metabolomic alterations at every time point, diurnal patterns were the most significant alterations seen, and coincided with the transitions from light to dark phases. Comparing ZT2 versus ZT14 remained relevant and provided data that was sufficient to claim altered gene expression (not oscillations) in a tissue specific manner in alignment with our other studies. Importantly, we have fully disclosed the different light dark cycles in our experiments in our methods section, and have added the following statement in the discussion to address this comment:

“It is important to note that the cecal microbial transfer experiments into germ-free mice (Figure 5) were on a slightly different light cycle (14:10) compared to all other studies (12:12), which was necessitated by our gnotobiotic facility standard operating procedures. It remains possible that this lighting change may alter some of the circadian and metabolic phenotypes under study.”

Further, the authors are drawing broad claims regarding circadian rhythmicity of microbiota as well as host parameters based on two timepoints in the conventionalized study, which is not acceptable. A minimum of 3 timepoints (at least to make the initial claim) are needed to draw conclusions regarding rhythms in gut microbes and host parameters. Suggest softening claims regarding these data as they relate to circadian dynamics.

We understand the reviewer’s concern and agree the term oscillation should be used correctly. In the revised version here, we only discuss circadian rhythmicity in datasets where at least 6 time points were collected (i.e. samples collected every 4 hours over a consecutive 24-hour period). In the introduction we mention oscillation when describing previous literature. In the results, serum parameters and gene expression data were analyzed every 4-hours for 24-hours and analyzed for oscillations by the cosinor method. Therefore, oscillations in the data could be interpreted. Initial cecal bacterial communities and serum metabolite analyses were also conducted every 4-hours for 24-hours. If experiments were performed at ZT2 and ZT14 we changed any reference oscillation regarding the data. It is important to note that anywhere we make a circadian claim in the paper, at least 6 times points have been collected in the same experiment.

7) Regarding the conventionalization study – can the authors provide statistics for the engraftment efficiency relative to the donors? The stacked bars from the donors are shown in panel f in Figure 2—figure supplement 1, but it does not seem like all animals colonized to an equal extent (i.e., conventionalized mice at ZT2 from the HFD CCTx group shown in Figure 5, panel b appear to be missing Blautia). For the mice that were collected at ZT2 vs. ZT14, were all the animals harvested from a single cage? Or were they spread out across cages? This is important for conventionalization, since colonization efficiency can be unique from cage to cage, creating artifacts within a timepoint if not accounted for.

We appreciate the reviewer’s concern regarding the engraftment efficiency of donor cecal microbial pools. Due to the oscillatory nature of the gut microbial community, the donor cecal pools used for engraftment only represent a snapshot in time of the microbial community. For this reason, it is not feasible to calculate an exact engraftment efficiency. However, we can determine the engraftment efficiency summed over two timepoints (ZT2 and ZT14) as % donor pool OTUs present in the recipient:

**Author response table 1. sa2table1:** 

Donor Cecal Pool	# Donor OTUs	# Matching Recipient OTUs (ZT2 + ZT14)	% Engraftment Efficiency
HFD	30	29	96.67
HFD + IMC	32	26	81.25

Additionally, the data represented in Figure 5, panel b are relative (not absolute) abundances and consequently, underrepresented OTUs may be excluded from visual representation at any one timepoint. Regarding the possibility of cage effect, all samples sequenced from ZT2 vs. ZT14 gnotobiotic mice were from animals housed in different cages, so can not be attributed to cage effects. We would like to thank the reviewer for bringing this critical point to our attention. We have updated the Methods to reflect this distinction.